# Towards practical and robust DNA-based data archiving using the yin–yang codec system

Zhi Ping [1,2,3,4,11], Shihong Chen [2,3,5,11], Guangyu Zhou[6,7,11], Xiaoluo Huang[1,11], Sha Joe Zhu[8], Haoling Zhang[1,2,3,4], Henry H. Lee[6], Zhaojun Lan[9], Jie Cui[2,3,5], Tai Chen[2,3,5], Wenwei Zhang[1,2], Huanming Yang[1,2,3], Xun Xu [1,2,4,5 ✉], George M. Church[3,6,10 ✉] and Yue Shen [1,2,3,4 ✉]

**DNA is a promising data storage medium due to its remarkable durability and space-efficient storage. Early bit-to-base transcoding schemes have primarily pursued information density, at the expense of introducing biocompatibility challenges or decoding failure. Here we propose a robust transcoding algorithm named the yin–yang codec, using two rules to encode two binary bits into one nucleotide, to generate DNA sequences that are highly compatible with synthesis and sequencing technologies. We encoded two representative file formats and stored them *in vitro* as 200 nt oligo pools and *in vivo* as a ~54 kbps DNA fragment in yeast cells. Sequencing results show that the yin–yang codec exhibits high robustness and reliability for a wide variety of data types, with an average recovery rate of 99.9% above $10^4$ molecule copies and an achieved recovery rate of 87.53% at $\leq 10^2$ copies. Additionally, the *in vivo* storage demonstration achieved an experimentally measured physical density close to the theoretical maximum.**

DNA is an ancient and efficient information carrier in living organisms. At present, it is thought to have great potential as an alternative storage medium because standard storage media can no longer meet the exponentially increasing data archiving demands. Compared with common information carriers, the DNA molecule exhibits multiple advantages, including extremely high storage density (estimated physical density of 455 EB per gram of DNA[1]), extraordinary durability (half-life >500 years (refs. [2,3])) and the capacity for cost-efficient information amplification.

Many strategies have been proposed for digital information storage using organic molecules, including DNA, oligopeptides and metabolomes[4–8]. Since current DNA sequencing technology has advantages in terms of both cost and throughput, storing digital information using DNA molecules remains the most well-accepted strategy. In this approach, the binary information from each file is transcoded directly into DNA sequences, which are synthesized and stored in the form of oligonucleotides or double-stranded DNA fragments *in vitro* or *in vivo*. Then, sequencing technology is used to retrieve the stored digital information. In addition, several different molecular strategies have been proposed to implement selective access to portions of the stored data, to improve the practicality and scalability of DNA data storage[9–11].

However, the use of basic transcoding rules (that is, converting [00, 01, 10, 11] to [A, C, G, T]) generates some specific patterns in DNA sequences that result in challenges regarding synthesis and sequencing[9,12,13]. For example, single-nucleotide repeats (homopolymers) longer than 5 nt might introduce a higher error rate during synthesis or sequencing[14,15]. Meanwhile, because of the nature of complementary base pairing (with A pairing to T and G to C),

DNA molecules may form structures such as hairpins or topological pseudoknots (i.e., secondary structure), which can be predicted by calculating the free energy from its sequence. It is reported that DNA sequences with stable secondary structure can be disadvantageous for sequencing or when using PCR for random access to and backup of stored information[16–19]. Additionally, DNA sequences with GC content <40% or >60% are often difficult to synthesize. Therefore, the length of homopolymers (in nt), the secondary structure (represented by the calculated free energy in kJ mol⁻¹) and the GC content (in %) are three primary parameters for evaluating the compatibility of coding schemes.

Previous studies on transcoding algorithm development have attempted to improve the compatibility of the generated DNA sequences. Early efforts, including those of Church et al. and Grass et al., introduced additional restrictions in the transcoding schemes to eliminate homopolymers, but this came at the expense of reduced information density[1,20,21]. Later studies pioneered other base conversion rules without compromising the information density. For example, the DNA Fountain algorithm adopted Luby transform codes to improve the information fidelity by introducing low redundancy as well as screening constraints on the length of homopolymers and the GC content while maintaining an information density of 1.57 bits nt⁻¹ (refs. [6,22]). However, the major drawback is the risk of unsuccessful decoding when dealing with particular binary features due to fundamental issues with Luby transform codes. This approach relies on the introduction of sufficient logical redundancy, that is, at the coding level, for error tolerance to ensure successful decoding. This is different from physical redundancy, which refers to the synthesis of excess DNA molecules, that is, increasing

[1]BGI-Shenzhen, Shenzhen, China. [2]Guangdong Provincial Key Laboratory of Genome Read and Write, BGI-Shenzhen, Shenzhen, China. [3]George Church Institute of Regenesis, BGI-Shenzhen, Shenzhen, China. [4]Shenzhen Institute of Synthetic Biology, Shenzhen Institutes of Advanced Technology, Chinese Academy of Sciences, Shenzhen, China. [5]China National GeneBank, BGI-Shenzhen, Shenzhen, China. [6]Department of Genetics, Harvard Medical School, Boston, MA, USA. [7]Department of Molecular and Cellular Biology, Harvard University, Cambridge, MA, USA. [8]Big Data Institute, Li Ka Shing Centre for Health Information and Discovery, University of Oxford, Oxford, UK. [9]School of Mathematical Science, Capital Normal University, Beijing, China. [10]Wyss Institute for Biologically Inspired Engineering, Harvard University, Boston, MA, USA. [11]These authors contributed equally: Zhi Ping, Shihong Chen, Guangyu Zhou and Xiaoluo Huang. ✉e-mail: xuxun@genomics.cn; gchurch@genetics.med.harvard.edu; shenyue@genomics.cn

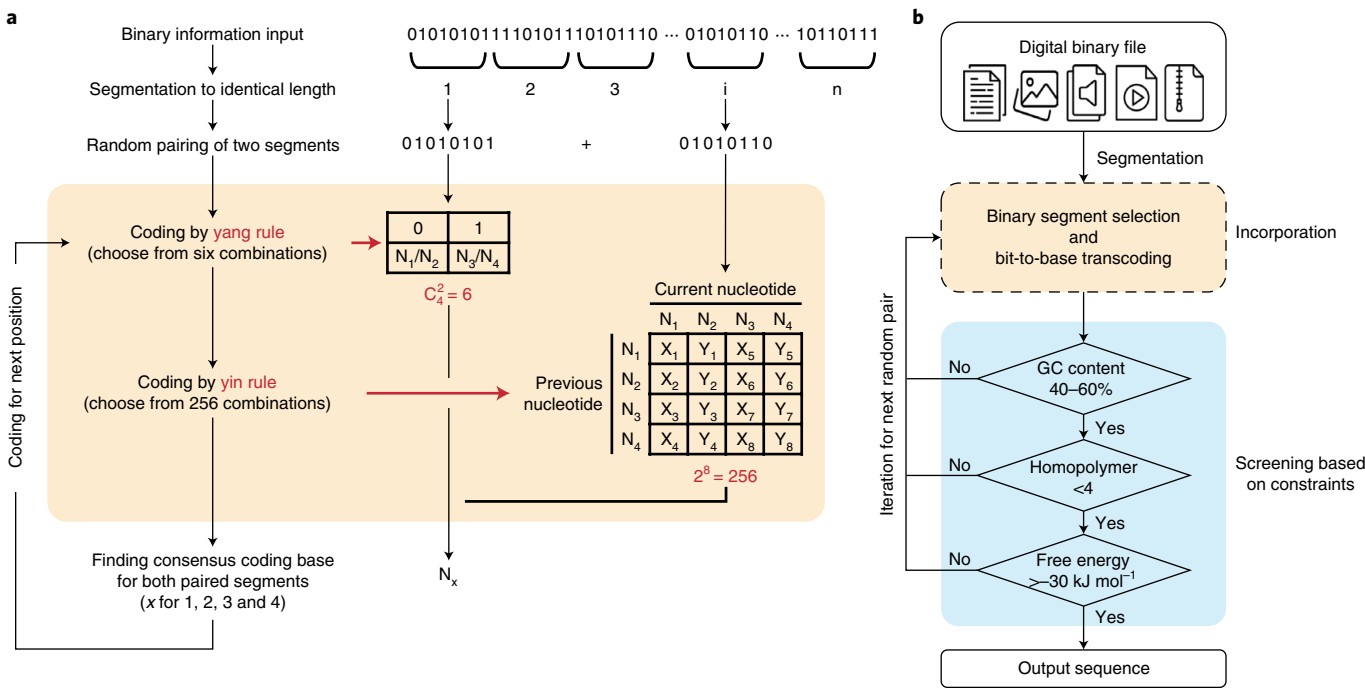

**Fig. 1 | Principles of the YYC. a**, The bit-to-base transcoding process of the YYC. N1, N2, N3 and N4 represent the nucleic acids A, T, C and G, respectively. $X_j$ and $Y_j$ represent different binary digits 0 and 1. When $j$ is an integer chosen from 1 to 8, $X_j + Y_j = 1$ and $X_j \times Y_j = 0$ (that is, eight independent sets of X and Y, with $X_j/Y_j$ being 1/0 or 0/1). $C_4^2$ means that the calculation for number of 2-combination in a set with 4 elements. **b**, A flowchart of the YYC encoding pipeline.

the copy number of DNA molecules for each coding sequence[23,24]. Reducing the logical redundancy could lead to a high probability of decoding failure, but excessive logical redundancy will decrease the information density and significantly increase the cost of synthesis[25]. Furthermore, specific binary patterns using these early algorithms may also create unsuitable DNA sequences, with either extreme GC content or long homopolymers (Supplementary Table 1). Therefore, developing a coding algorithm that can achieve high information density but, more importantly, perform robust and reliable transcoding for a wide variety of data types in a cost-effective manner is necessary for the development of DNA-based information storage in practical applications[25–27].

To achieve this goal, we propose herein the yin–yang codec (YYC) coding algorithm, inspired from the traditional Chinese concept of yin and yang, representing two different but complementary and interdependent rules, and we demonstrate its performance by simulation and experimental validation. The advantage of the YYC is that the incorporation of the yin and yang rules finally leads to 1,536 coding schemes that can suit diverse data types. We demonstrate that YYC can effectively eliminate the generation of long homopolymer sequences while keeping the GC content of the generated DNA sequences within acceptable levels. Two representative file formats (.jpg and .txt) were chosen for storage as oligo pools *in vitro* and a 54 kbps DNA fragment *in vivo* in yeast cells to evaluate the robustness of data recovery. The results show that YYC exhibits good performance for reliable data storage as well as physical density reaching the scale of EB per gram.

## Results

**The general principle and features of the YYC.** In nature, DNA usually exists in a double-stranded structure. In some organisms such as phages, both strands encode genetic information to make the genome more compact. Inspired by this natural phenomenon, we used the basic theory of combinatorics and cryptography to develop a codec algorithm on the basis of Goldman's rotating encoding strategy[28,29]. Unlike other coding schemes developed

using fixed mapping rules, the YYC provides dynamic combinatory coding schemes and can thus generate optimal DNA sequences to address the DNA synthesis and sequencing difficulties found when generating DNA sequences with long homopolymers, extreme GC content or complex secondary structure.

The general principle of the YYC algorithm is to incorporate two independent encoding rules, called 'yin' and 'yang', into one DNA sequence (called 'incorporation'), thereby compressing two bits into one nucleotide (Fig. 1a). Here, we use N1, N2, N3 and N4 to represent the four nucleic acids A, T, C and G, respectively. For one selected combinatory coding scheme, an output DNA sequence is generated by the incorporation of two binary segments of identical length. In the first step, the yang rule is applied to generate six different coding combinations. Then, in the yin rule, N1 and N2 are mapped to different binary digits, while N3 and N4 are also mapped to different binary digits independent of N1 and N2, leading to a total of 256 different coding combinations. Application of the yin and yang rules at one position will yield one and only one consensus nucleotide (Supplementary Fig. 1 and Supplementary Video 1). Meanwhile, according to the four different options for the previous nucleotide, the two groups (N1/N2 and N3/N4) also have independent options for the mapping to 0 and 1. Therefore, the incorporated yin and yang rules provide a total of 1,536 (6 × 256) combinations of transcoding schemes to encode the binary sequence. More details are described in the Supplementary information.

To demonstrate the compatibility of the YYC algorithm and quantify its featured parameters in comparison with other early DNA-based data storage coding schemes, the 1 GB data collection was transcoded by using the YYC as well as other early coding algorithms for comparison[1,20–22]. As shown in Table 1, the flexible screening process introduced after the incorporation of binary segments for both the YYC and DNA Fountain algorithms provides more possibilities for obtaining DNA sequences with desired GC content values between 40% and 60%. Like all the other coding algorithms, the YYC also introduces constraints to set the maximum homopolymer length at 4, considering computing resources

**Table 1 | Comparison of DNA-based data storage schemes**

| | | Church et al. | Goldman et al. | Grass et al. | Erlich et al. | Chen et al. | This work (YYC) |
|---|---|---|---|---|---|---|---|
| General attributes | Error correction strategy | No | Repetition | RS | Fountain | LDPC | RS |
| | Robustness against excessive errors | Yes | Yes | Yes | No | Yes | Yes |
| | Information density (bits nt$^{-1}$)[a] | 1[a] | 1.58[a] | 1.78[a] | 1.98[a] | 1.24[b] | 1.95 |
| | Physical density achieved (Ebytes g$^{-1}$) — In vitro | 0.001[a] | 0.002[a] | 0.025 | 0.21[a] | N/A[b] | 2.25 |
| | Physical density achieved (Ebytes g$^{-1}$) — In vivo | N/A | N/A | N/A | N/A | 270.7[b] | 432.2[b] |
| Biotechnical compatibility | GC content (%) of sequences | 2.5–100 | 22.5–82.5 | 12.5–100 | 40–60 | N/A | 40–60 |
| | Maximum homopolymer length (nt) | 3 | 1 | 3 | 4 | N/A | 4 |
| | Ratio (%) of sequences with free energy >−30 kJ mol$^{-1}$ | 71.72 | 25.87 | 90.14 | 65.25 | N/A | 100 |

The schemes are presented chronologically based on publication date. The biotechnical compatibility is obtained according to in silico simulation of 1 GB file collections (Methods). LDPC, low-density parity check. [a]Information based on data from ref. [22]. [b]Calculated value in the form of data coding in a DNA fragment integrated into the yeast genome. N/A means the data is not available in the corresponding studies.

as well as the technical limitations of DNA synthesis and sequencing. In addition, the YYC considers the secondary structure of the generated DNA sequences as part of the compatibility analysis, by rejecting all DNA sequences with free energy lower than −30 kcal mol$^{-1}$. In addition, the statistics of other features were analysed using the data collection and several test files in various data formats (Supplementary Figs. 2, 3 and 4 and Supplementary Tables 2, 3 and 4), suggesting that the YYC has no specific preference regarding the data structure and maintains a relatively high level of information density, ranging from 1.75 to 1.78 bits per base (Methods and Supplementary Fig. 2). For some cases, our simulation analysis suggests that a few (approximately seven) coding schemes from the collection of 1,536 might generate DNA sequences with identity between 80% and 91.85% (Supplementary Fig. 3), but at very low frequency.

Given these results, YYC offers the opportunity to generate DNA sequences that are highly amenable to both the 'writing' (synthesis) and 'reading' (sequencing) processes while maintaining a relatively high information density. This is crucially important for improving the practicality and robustness of DNA data storage. The DNA Fountain and YYC algorithms are the only two known coding schemes that combine transcoding rules and screening into a single process to ensure that the generated DNA sequences meet the biochemical constraints. The comparison hereinafter thus focuses on the YYC and DNA Fountain algorithms because of the similarity in their coding strategies.

**In silico robustness analysis of YYC for stored data recovery.** The robustness of data storage in DNA is primarily affected by errors introduced during 'writing' and 'reading'. There are two main types of errors: random and systematic errors. Random errors are often introduced by synthesis or sequencing errors in a few DNA molecules and can be redressed by mutual correction using an increased sequencing depth. Systematic errors refer to mutations observed in all DNA molecules, including insertions, deletions and substitutions, which are introduced during synthesis and PCR amplification (referred to as common errors), or the loss of partial DNA molecules. In contrast to substitutions (single-nucleotide variations, SNVs), insertions and deletions (indels) change the length of the DNA sequence encoding the data and thus introduce challenges regarding the decoding process. In general, it is difficult to correct systematic errors, and thus they will lead to the loss of stored binary information to varying degrees.

To test the robustness baseline of the YYC against systematic errors, we randomly introduced the three most commonly seen errors into the DNA sequences at a average rate ranging from 0.01% to 1% and analysed the corresponding data recovery rate in comparison with the most well-recognized coding scheme (DNA Fountain) without introducing an error correction mechanism. The results show that, in the presence of either indels (Fig. 2a) or SNVs (Fig. 2b), YYC exhibits better data recovery performance in comparison with DNA Fountain, with the data recovery rate remaining fairly steady at a level above 98%. This difference between the DNA Fountain and other algorithms, including YYC, occurs because uncorrectable errors can affect the retrieval of other data packets through error propagation when using the DNA Fountain algorithm. Although the robustness to systematic errors can be improved by introducing error correction codes, such as the Reed–Solomon (RS) code or low-density parity-check code[21,30,31], when the error rate exceeds the capability of such codes, the error correction will fail to function as designed. Furthermore, it is universally acknowledged that no efficient error correction strategies have been experimentally verified to be effective for insertions and deletions[32], let alone loss of the entire segment coding sequence. Therefore, in real applications, traditional error correction codes might play a limited role for improving robustness because of their inability to correct indels or the loss of the entire sequence.

As the other major factor for data recovery, the loss of partial DNA molecules can also affect the success rate of data retrieval[33]. Like early coding schemes (for example, those of Church et al., Goldman et al. and Grass et al.), the YYC is also designed like a linear block nonerasure code, with a linear relationship between data loss and the encoded sequence loss. Nevertheless, because of the convolutional binary incorporation of YYC, errors that cannot be corrected within one DNA sequence will lead to the loss of information for two binary sequences. In contrast, the DNA Fountain algorithm uses a different data retrieval strategy based on its grid-like topology of data segments, and theoretically, its data recovery cannot be guaranteed when a certain number of DNA sequences are missing[22]. In this work, in silico simulation of the data recovery rate in the context of a gradient of DNA sequence loss was performed. The results show that the YYC exhibits linear retrieval, as predicted. The data recovery percentage remains at 98% when the sequence loss rate is <2%. Even with 10% sequence loss, the YYC can recover the remaining ~90% of the data. In contrast, when the sequence loss rate exceeds 1.7%, the data recovery rate of the DNA

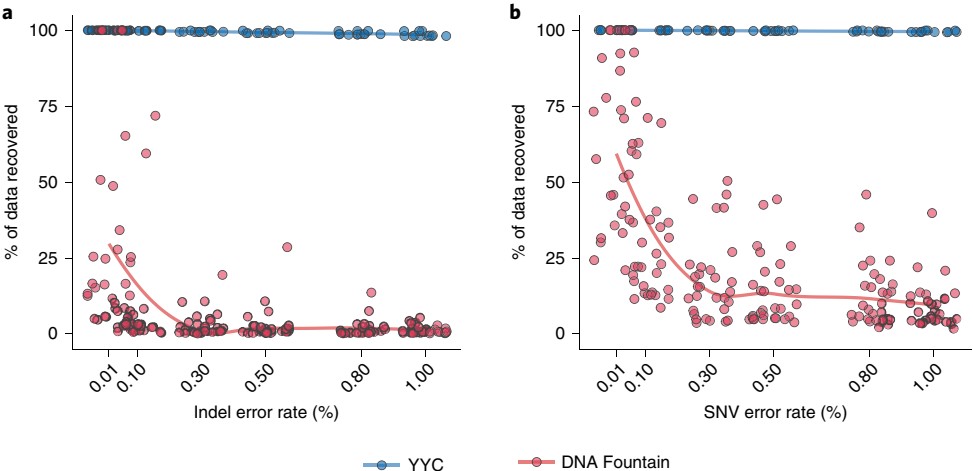

**Fig. 2 | Robustness analysis for the YYC and DNA Fountain coding schemes. a,b**, The binary data recovery rate of the YYC (blue) and DNA Fountain (red) coding strategies without any error-correction algorithm for indels (**a**) and SNVs (**b**) introduced randomly with an error rate of 0.01%, 0.1%, 0.3% 0.5%, 0.8% or 1%.

Fountain algorithm becomes highly volatile and drops significantly (Supplementary Table 5). Fountain codes function well for telecommunications and internet communications because the information transfer and verification are synchronous, thus giving the information source a chance to send more data packets for successful data recovery. However, the information writing (synthesis) and reading (sequencing) processes for DNA-based data storage are heterochronic, meaning that multiple, stepwise molecular manipulations are involved during the whole process. This makes the immediate transmission of additional data packets unrealistic for DNA-based data storage. Thus, although rateless codes including Fountain codes may improve the performance by adjusting their configuration and parameters, such coding schemes that suffer from the risk of uncertain decodability are not ideal for DNA-based data storage applications.

**Experimental validation of the YYC with *in vitro* storage.** To determine the compatibility of the YYC with current biochemical technologies, including DNA synthesis, PCR amplification and sequencing, we encoded three digital files (two text files, one each in English and Chinese, and an image) using the YYC and stored the encoded file in the form of 10,103 200 nt oligos *in vitro*. The sequence design of the oligos generated by the YYC transcoding is illustrated in Fig. 3.

Three oligo pools were synthesized for an experimental validation of *in vitro* storage. Pool 1 (P1) includes oligos with 25% logical redundancy. In comparison, two independent oligo pools (P2 and P3) of these three files, both transcoded by the DNA Fountain algorithm, were also synthesized using previously described settings (Fig. 3a):[22] Pool 2 (P2) includes 10,976 oligos encoding these three files individually, where it has been reported previously that logical redundancy is required for successful decoding, while pool 3 (P3) encodes the same files in a .tar archiving compressed package. The RS error-correction code was used in all three oligo pools.

The average molecule copy (AMC) number of the P1, P2 and P3 master pools is estimated to be ~10[7]. A ten-fold serial dilution of P1, P2 and P3, with estimated AMC number from 10[6] to 10[0] for each oligo pool, was performed for sequencing to evaluate the minimal copy number of oligos required for successful file retrieval, as well as the robustness performance against DNA molecule loss (Fig. 3b). The sequencing results demonstrate that ~99.9% of the corresponding data from P1 can be recovered at AMC numbers above 10[3], with no preference regarding the specific data format (Fig. 3b and

Supplementary Fig. 5c). As the AMC number decreases in magnitude, the decoding robustness shows an increase of instability. The average data recovery rate decreases to 71.2% at an AMC number of 10[2], ranging from 65.69% to 87.53% for each stored file. It drops further to below 10% when the AMC number is less than 10[1]. In general, the YYC exhibits linear retrieval trend, which is positively correlated with the amount of data-encoding DNA molecules retained (Fig. 3c). For the DNA Fountain algorithm, the data recovery rate at an AMC number above 10[4] is comparable to that of the YYC, but it drops significantly at lower AMC numbers from 10[3] to the single-copy level (Fig. 3b). Especially for P3, the data was first .tar archived and then transcoded for storage. According to our experimental results, a maximum of 32.83% of the data package can be retrieved at lower levels of AMC number (Supplementary Data 1). However, the disruption of the compressed package leads to total loss of the original data. In addition, it has been suggested previously that most random errors introduced during synthesis or sequencing can be corrected by increasing the sequencing depth[34]. However, we found that, although lost sequences could be retrieved by such deep sequencing (Supplementary Fig. 6a), these sequences are at relatively low depth and contain more errors (Supplementary Fig. 6b). Therefore, such retrieved sequences are insufficient for valid information recovery. The current results suggest that loss of DNA molecules is the major factor affecting the data recovery rate, and that even high sequencing depth cannot improve the recovery rate if a certain amount of data-encoding DNA molecules are lost. In general, the relationship found between the information recovery rate and the sequence retention rate of each synthesized oligo pool in the *in vitro* experiment is consistent with that found in the in silico simulations, for the YYC and DNA Fountain algorithms.

To further investigate the compatibility of the coding schemes for different binary patterns from various files, we examined the performance of the YYC and DNA Fountain algorithms on test files in various formats. It is reported that information loss and decoding failure in DNA data storage can also result from original defects in the transcoding algorithms[26,35–38]. Therefore, increasing the logical redundancy could greatly improve the probability of successful decoding for all the coding schemes. However, too much logical redundancy requires the synthesis of more nucleotides and thus reduces the information density. Therefore, it is very important to keep the logical redundancy level in a controllable range for massive file archiving. Based on the transcoding simulations for these files, it is suggested that, especially for nonexecutable files, the

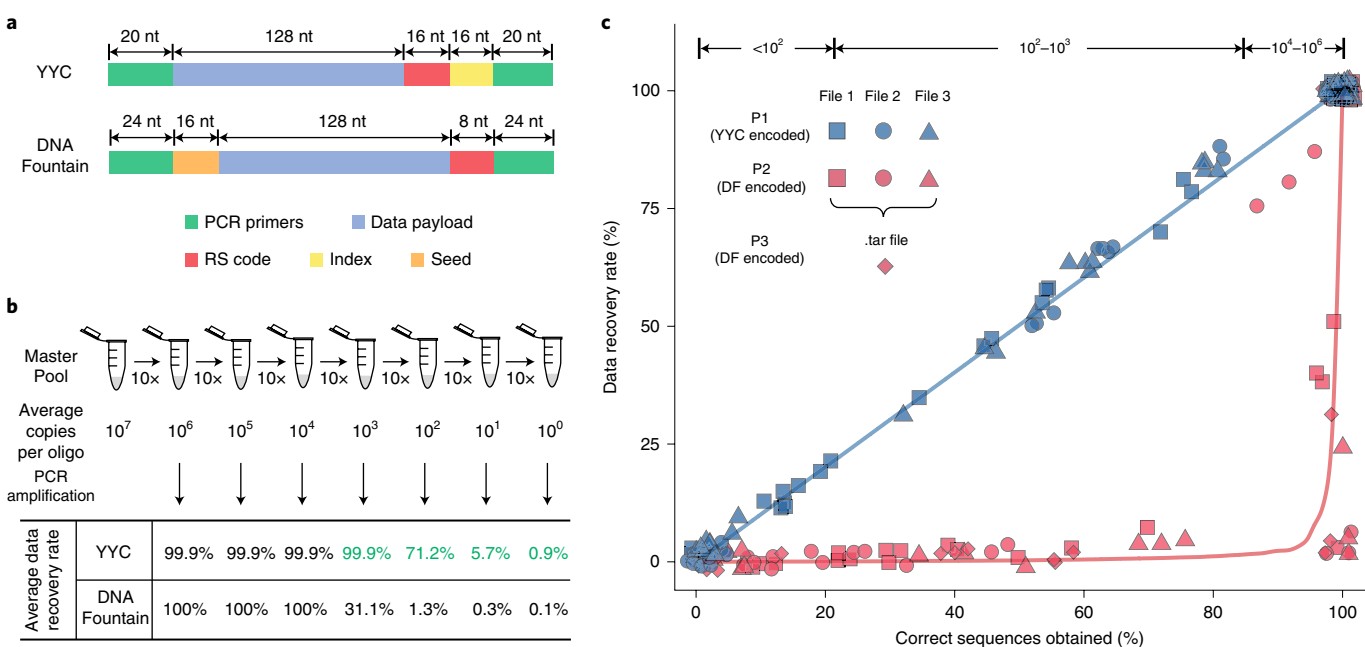

**Fig. 3 | Experimental validation of *in vitro* binary data storage using the YYC and DNA Fountain coding strategies. a**, The sequence design of the 200 nt oligo generated by the YYC and DNA Fountain algorithms for *in vitro* data storage. **b**, The serial dilution experiment of the synthesized oligo pool. The average copy number of each oligo sequence is calculated accordingly to the original oligo pool. The average data recovery rates are calculated based on the sequencing result of the PCR products of the diluted samples (the data recovery rates of YYC samples with low molecule copy number (≤10³) is labeled with green color; the DNA molecule copy number for each sample after the PCR amplification exceeds 10⁸). **c**, Analysis of the YYC and DNA Fountain (abbreviated as DF) algorithms by sequencing of corresponding diluted samples and calculation of the data recovery rate of each file encoded by YYC and DNA Fountain at the corresponding oligo copy number.

DNA Fountain algorithm exhibits variable requirements for the level of logical redundancy, leading to a varying information density (Supplementary Table 6). In contrast, the YYC coding scheme always requires a relatively low level of logical redundancy, resulting in more general compatibility with a broader range of file types and demonstrating a more stable information density.

**Experimental validation of the YYC with *in vivo* storage.** *In vivo* DNA data storage has attracted attention in recent years because of its potential to enable economical write-once encoding with stable replication for multiple data retrievals[30]. However, whether and the extent to which the robustness of a coding scheme can be maintained against spontaneous mutations or unexpected variations accumulated during long-term passaging of living cells has not been comprehensively investigated previously. Thus, we encoded a portion of a text file (Shakespeare Sonnet.txt) into a 54,240 bp DNA fragment containing 113 data blocks using the YYC and evaluated its potential data robustness for *in vivo* DNA data storage applications. The sequence design for each data block included a 456 bp data payload region and a 24 nt RS code region (Fig. 4a). The generated DNA fragment was first synthesized de novo into 60 of ~1 kbps subfragments and then assembled into 20 of ~2.8 kbps fragments (Methods). Taking advantage of the high homologous recombination efficiency of yeast, these fragments were directly transformed into yeast strain BY4741 together with the linearized low-copy centromeric vector pRS416 to enable one-step full-length DNA assembly *in vivo*. After ~1,000 generations by batch transfer of cell culture, we evaluated the robustness of the YYC scheme by subjecting 15 single colonies to whole-genome sequencing (Fig. 4b). First, in addition to indels or SNVs that could be introduced during construction or passaging of the cells, we also observed varying degrees of partial fragment loss from ~21.1 kbps to ~51.4 kbps among all 15 selected single colonies, leading to different levels of data recovery

from 38.9% to 95.0% (Supplementary Table 7 and Fig. 4c,d). Since the observed indels or large deletions might lead to frameshifts of the data-encoding DNA sequence and subsequent decoding failure, a single-winner plurality voting strategy was applied to generate a consensus sequence from the reconstruction and alignment of multiple colonies (Fig. 4c). By doing so, we reconstructed a full sequence with 66 SNVs that cannot be corrected by the RS code introduced into the data block and fully recovered the stored data. In addition, to test the maximum physical density achievable in this study, we further integrated the constructed data-encoding DNA fragment into chromosome II of the yeast BY4741 genome. Therefore, for each resulting yeast cell, the data-encoding DNA is maintained at one single-copy level. By doing so, we successfully demonstrated that a physical density of ~432.2 EB g⁻¹ can be achieved, suggesting a significant increase by three orders of magnitude than that demonstrated in prior work[22,39,40] (Table 1).

## Discussion
The YYC transcoding algorithm offers several advantages. First, it successfully balances high robustness, compatibility and a considerable information density for DNA data storage compared with other early efforts. With the gradual popularization of DNA data storage, it is crucially important that the developed coding algorithms can perform robust and reliable transcoding for a wide variety of data types, especially for data with specific binary patterns. Before transcoding, compression algorithms such as Lempel–Ziv–Welch, Gzip or run-length encoding can be used to make the byte frequency[41] more balanced and avoid specific data patterns (Supplementary Table 1), thus improving the compatibility of the generated DNA sequences. However, because compression will change the original information structure, our results show that even partial loss of the DNA molecules will result in total failure to recover the compressed data. Current compression algorithms are not designed for DNA

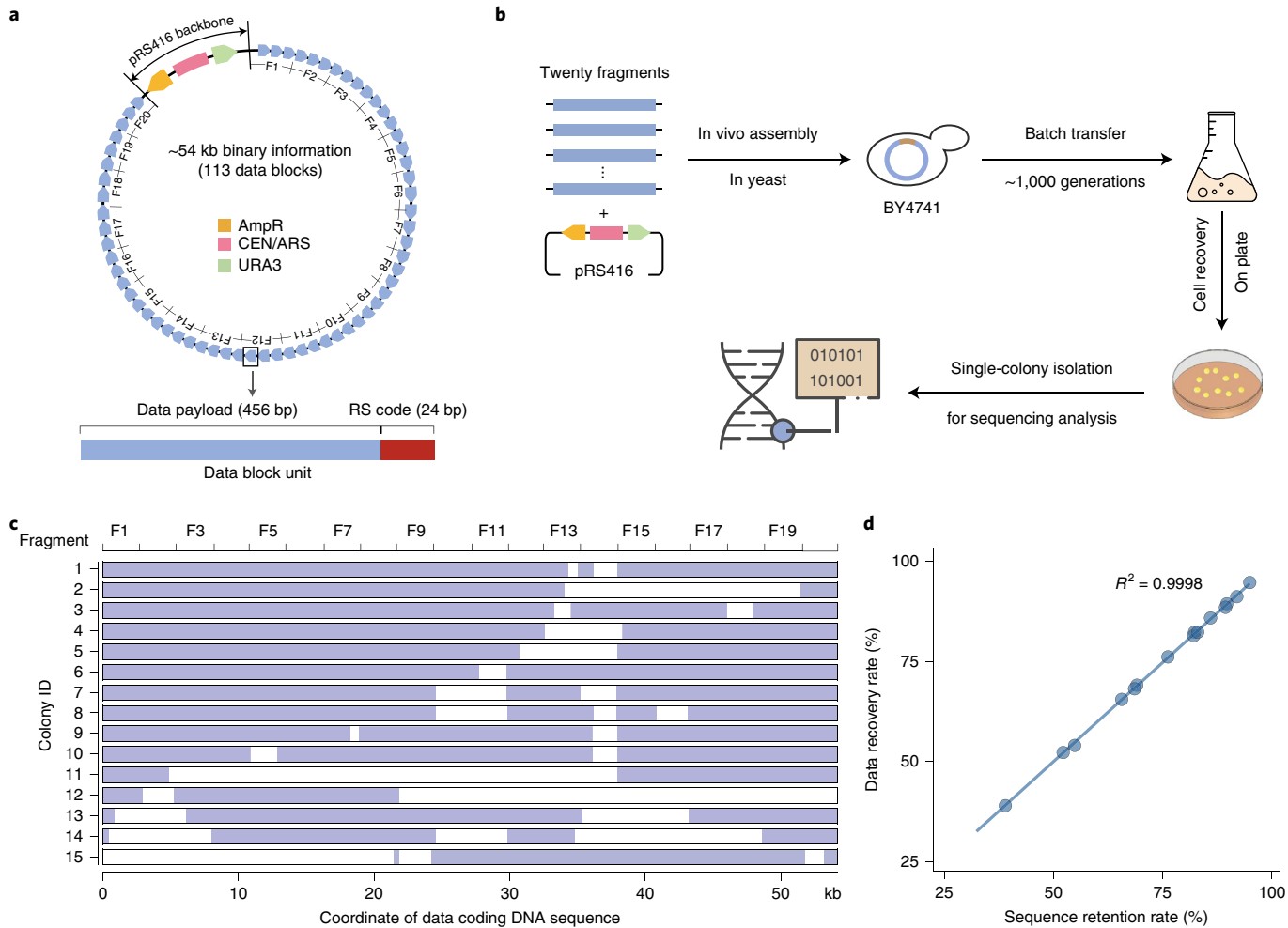

**Fig. 4 | In vivo experimental validation. a**, The design of the DNA sequence generated by the YYC for *in vivo* data storage in the form of a ~54 kbps data-encoding DNA fragment in the pRS416 vector. AmpR: ampicillin resistance gene; CEN: yeast centromere sequence; ARS: autonomously replicating sequence; URA3: a selective marker gene. **b**, The workflow of the data recovery analysis for *in vivo* data storage after batch transfer of ~1,000 generations. **c**, The fate of the ~54 kbps data-coding fragment in each selected yeast strain is indicated as preserved (light blue) or deleted (white). **d**, The correlation between the retention rate of the data-encoding DNA fragment and the data recovery in each selected yeast strain (dots). $R^2$ indicates the coefficient of determination. Solid line is the fitted curve obtained using the command ggplot:geom_point (), geom_smooth (method="lm", se=FALSE).

data storage, and further refinement can be performed to compress data appropriately for robust bit-to-base transcoding. Another potential advantage of the YYC is the flexibility of the rule incorporation from its 1,536 options. Considering broader application scenarios, the YYC offers the opportunity to incorporate multiple coding schemes for the transcoding of a single file, thus providing an alternative strategy for secure data archiving. Furthermore, coding schemes for DNA storage can be modularized into data transcoding, assignment of indices, error correction, redundancy, etc., thus providing more options to be combined freely be users. In our early work, we also demonstrated an integration system called 'Chamaeleo' in which the YYC could be used compatibly for bit-to-base encoding together with other modules[42]. Further optimization and functionalities can be incorporated into the system as well.

Our sequencing results show that the error rate of the synthesized oligo pools is ~1%, and in addition, ~1.2% of oligos are lost when mapping with the designed oligo sequence collections. There are two main steps in which systematic errors could be introduced: the synthesis of data-encoded oligos and PCR amplification to obtain a sufficient amount of DNA for sequencing. In general, random errors introduced during sequencing can be corrected easily by using a sufficient sequencing depth, but errors introduced during

PCR amplification can be problematic. Error-correction codes can improve the information retrieval, but logically redundant sequences including both inner and outer codes can play a more important role in retrieving lost sequences and correcting errors for reliable DNA-based data storage. The length of the DNA sequence may also limit the pool capacity of a DNA storage system. These issues could be addressed in the future by using DNA synthesis technology with high stepwise efficiency, throughput and fidelity, which could yield longer DNA sequences and a high quantity of DNA and avoid amplification. For the demonstration of *in vivo* DNA storage, we find that there are random 1 nt indels and deletions of varying sizes in different data-coding regions across the selected single colonies, which could cause issues with data stability and recovery after long-term storage. Hence, it is critical that the coding strategy used should be able to retrieve as much information as possible. In addition, we also demonstrate herein that applying a voting strategy on a population of cells can further increase the possibility of fully recovering the stored information. Nevertheless, future efforts to improve the stability of exogenous artificial DNA in host cells is necessary to avoid unexpected information loss during passaging. The theoretical information density of DNA storage of 2 bits per base cannot be attained in real applications due to the setting of indices, the

error-correction strategy, intrinsic biochemical constraints and the technical limitations of the DNA synthesis and sequencing procedures[24]. The introduction of 'pseudobinary' segments in our study will also reduce the information density. Nevertheless, compared with another recent study on data storage using artificial yeast chromosomes in living yeast cells, the current results indicate better performance in terms of information density[30].

## Methods

**The YYC strategy.** *Demonstration of the YYC transcoding principle.* In the example referred to as coding scheme no. 888 in Supplementary Fig. 1a, the yang rule states that [A, T] represents the binary digit 0 while [G, C] represents the binary digit 1. Meanwhile, the yin rule states that the local nucleotide (the current nucleotide to be encoded) is represented by the incorporation of the previous nucleotide (or 'supporting nucleotide') and the corresponding binary digit (Supplementary Fig. 1b). During transcoding, these two rules are applied respectively for two independent binary segments and transcoded into one unique DNA sequence, while decoding occurs in the reverse order. For example, given an input signal formed of 'a' and 'b' of '10110011' and '01011101', respectively, the transcoding scheme will start with the first nucleotide in each segment. According to the yang rule, '1' in 'a' provides two options [C, G]. With the predefined virtual nucleotide in position 0 as 'A', the yin rule and '0' for 'b' also provide two options [A, G] (Supplementary Fig. 1a). Therefore, the intersection of these two sets generates the unique base [G] transcoding the first binary digit in these two segments. Similarly, the rest of the two segments can be converted into a unique nucleotide sequence (Supplementary Fig. 1b and Supplementary Video 1). Note that switching the binary segments will change the transcoded result, which means that [a: Yang, b: Yin] and [b: Yang, a: Yin] will result in the generation of completely different DNA sequences. Generally, only one, fixed incorporated coding scheme is selected to transcode each dataset. Nevertheless, multiple coding schemes can be used for transcoding in encryption applications, where the corresponding information describing the coding schemes used would be stored separately.

*Incorporation of the YYC transcoding pipeline.* Considering the features of the incorporation algorithm, binary segments containing excessively imbalanced 0s or 1s will tend to produce DNA sequences with extreme GC content or undesired repeats. Therefore, binary segments containing a high ratio of 0 or 1 (>80%) will be collected into a separate pool and then selected to incorporate with randomly selected binary segments with normal 0-to-1 ratios.

*Constraint settings of the YYC transcoding screening.* In this study, a working scheme named 'YYC-screener' is established to select valid DNA sequences. By default, the generated DNA sequences (normally ~200 nt) with a GC content >60% or >40%, carrying >6-mer homopolymer regions or possessing a predicted secondary structure of <−30 kcal mol⁻¹ are rejected. Then, a new run of segment pairing will be performed to repeat the screening process until the generated DNA sequence meets all the screening criteria. Considering that DNA sequencing and synthesis technologies continue to evolve rapidly, the constraint settings are designed as nonfixed features to allow user customization. In this work, the constraints are set as follows: a GC content between 40% and 60%, a maximum homopolymer length <5 and a free energy $\geq$−30 kcal mol⁻¹ (the free energy of the secondary structure is calculated using Vienna RNA version 2.4.6).

**In silico transcoding simulation.** *Computing and software.* All encoding, decoding and error analysis experiments were performed in an Ubuntu 16.04.7 environment running on an i7 central processing unit with 16 GB of random-access memory using Python 3.7.3.

*Input files and parameters for simulation.* The test files included 113 journal articles (including images and text), 112 .mp3 audio files from *Scientific American* and the supplementary video files from 33 journal articles.

To compare the compatibility of the different coding schemes, all the test files were transcoded by using Church's code, Goldman's code, Grass' code, DNA Fountain and the YYC in the integrated transcoding platform 'Chamaeleo' that we developed[42]. The segment length of the binary information was set as 32 bytes. For Church's code, Goldman's code and Grass' code, the original settings as previously reported were used in this study. For the DNA Fountain and YYC algorithms, the constraints were set as follows: a GC content of 40–60% and a maximum allowed homopolymer length of 4. For the free energy constraint in the YYC algorithm, the cutoff for probe design was set as −13 kcal mol⁻¹ for a ~20 nt DNA sequence, and considering a length of the data-coding DNA of 160 nt, we adjusted the cutoff to −30 kcal mol⁻¹ (refs. [43,44]).

Additional transcoding simulation tests were performed to evaluate the robustness and compatibility of the DNA Fountain and YYC algorithms. The DNA Fountain source code was used to perform encoding and decoding tests on nine different file formats and ten bitmap images with the default parameter settings (c-dist = 0.025, delta = 0.001, header size = 4, homopolymer = 4, GC = 40–60%)

with minimum decodable redundancy. The oligo length of both strategies was set as 152 bases with indices or seeds for data retrieval and without error-correction codes. To determine the minimum redundancy required for file decoding, a test interval of minimum redundancy was set as 1%, and the maximum redundancy allowed was 300%. In some cases, the process terminated with a system error, which might be caused by stack overflow.

**Experimental validation.** *File encoding using the YYC and DNA Fountain algorithms.* The binary forms of three selected files (9.26 × 10⁵ bits, 7.95 × 10⁵ bits and 2.95 × 10⁵ bits) were extracted and segmented into three independent 128 bit segment pools. A 16 bit RS code was included to allow the correction of up to two substitution errors introduced during the experiment. Next, four 144 bit binary segments (data payload + RS code) were used to generate a fifth redundant binary segment to increase the logical redundancy. Then, another 16 bit index was added into each binary segment to infer its address in the digital file and in the oligo mixture for decoding. Coding scheme no. 888 from the YYC algorithm was applied to convert the binary information into DNA bases. The aforementioned 'YYC-screener' was used to select viable DNA sequences. Eventually, 8,087 of 160 nt DNA sequence segments were generated. To allow random access to each file, a pair of well-designed 20 nt flanking sequences were added at both ends of each DNA sequence. Finally, an oligo pool containing 10,103 single-stranded 200 nt DNA sequences was obtained.

For DNA Fountain, the recommended default settings from its original report[22] (c-dist = 0.1, delta = 0.5, header size = 4, homopolymer = 4, GC = 40–60%), with the exception of redundancy, were used to generate the DNA oligo libraries. The minimum redundancy to ensure successful decoding was determined. Therefore, 13%, 22%, 73% and 12% logical redundancy was added for a .tar archiving compressed file, text1, text2 and image files, respectively. Finally, an oligo library encoding a .tar archiving compressed file (9,185 sequences) and an oligo library encoding the mixed three individual files (10,976 sequences) were obtained.

A part of one text file (~13 kB), was transcoded into DNA sequences by YYC for *in vivo* storage using a similar procedure, but the binary segment length was set as 87 bytes (or 456 bits). As described in the main text, the sequence was divided into 113 data blocks of 456 nt each. To increase the fidelity, a 24 nt RS code was added. The total data payload region as double-stranded DNA for *in vivo* storage is (456 + 24) × 113 = 54,240 bp.

*Synthesis and assembly.* The three oligo pools were outsourced for synthesis by Twist Biosciences and delivered in the form of DNA powder for sequencing.

For *in vivo* storage, the 54,240 bp DNA fragment was first segmented into 20 subfragments (2,500–2,900 bp) with overlapping regions and then further segmented into building blocks (800–1,000 bp, hereafter referred to as blocks). For each block, 20 of 80-nt oligos were synthesized with a commercial DNA synthesizer (Dr. Oligo, Biolytic Lab Performance) and then assembled into blocks by applying the polymerase cycling assembly method using Q5 High-Fidelity DNA Polymerase (M0491L, NEB) and cloned into an accepting vector for Sanger sequencing. Then, the sequencing-verified blocks were released from their corresponding accepting vector by enzymatic digestion for the assembly of subfragments by overlap extension (OE-)PCR. Gel purification (QIAquick gel extraction kit, 28706, QIAGEN) was performed to obtain the assembled subfragments. By transforming all 20 subfragments (300 ng each) and the low-copy accepting vector pRS416 into BY4741 yeast using LiOAc transformation[45] and taking advantage of yeast's native homologous recombination, the full-length ~54 kb DNA fragment was obtained. After 2 days of incubation on selective media (SC-URA, 630314, Clontech) at 30 °C, 16 single colonies were isolated for liquid culturing in YPD (Y1500, Sigma) before sequencing. One of the colonies showed very low target region coverage and was excluded from further analysis.

For the *in vivo* storage demonstration via genome integration, the full assembled fragment was inserted right after gene *YBR150C* on chromosome II with the LEU2 marker for selection via yeast transformation. The transformants were recovered on SC-Leu plates (SC-LEU, 630310, Clontech). Three positive colonies were isolated for genomic DNA extraction and sequencing.

*Library preparation and sequencing.* For library preparation of the synthesized oligo pool, the DNA powder was first dissolved in double-distilled water (ddH₂O) to obtain a standard solution, with an average of 10⁷ molecules µL⁻¹ per oligo for each synthesized oligo pool. Then, the standard solution was serially diluted by 10-fold to create the seven working solutions (WSs) of WS6 to WS0 with average concentration of 10⁶ to 10⁰ DNA molecules µL⁻¹, respectively, for each oligo pool. Then, each WS was amplified by PCR with three technical replicates to obtain the amplified product for P2 and each of the three different files for P1 and P3. PCR amplification was performed using 25 µL 2× Q5 High-Fidelity DNA Polymerase master mix (M0491L, NEB), 2 µL forward and reverse primer pairs each (10 µM each), 1 µL template DNA and 20 µL ddH₂O added to a final reaction volume of 50 µL. To obtain a sufficient amount of product for later sequencing, the PCR thermal cycler programme settings for P1 and P3 were as follows: 98 °C for 5 min; 23, 27, 32, 36, 40, 44 and 48 cycles of 98 °C for 10 s, 62 °C for 15 s and 72 °C for 10 s; and final extension at 72 °C for 2 min. The PCR settings for P2 were almost the same, but the annealing temperature was 60 °C for DF-F1 (Forward

primer 1) and 58 °C for DF-F2 (Forward primer 2) and DF-F3 (Forward primer 3). The concentrations of products were measured using gel electrophoresis and Qubit fluorometer, and corresponding molecules per microlitre values were also calculated (Supplementary Data 1). All amplified DNA libraries were then sequenced using DIPSEQ-T7 sequencing[46].

For *in vivo* storage, the methods for genomic DNA extraction and standard library preparation of the yeast colonies were described in previous studies[47]. The prepared samples were sequenced using the DNBSEQ-G400 (MGISEQ-2000) and DNBelab sequencing platform[48].

*Data analysis.* In total, >3 G PE-150 reads were generated for the *in vitro* storage experimental validation. Sequencing data with an average depth of 100× were randomly subsampled for information retrieval. The reads were first clustered and assembled to complete sequences for each type of oligo. Flanking primer regions were removed, DNA sequences were decoded to binary segments using the reverse operation of encoding and substitution errors were corrected using the RS code. The binary segments were reordered according to the address region. During this process, 'pseudobinary' segments were removed based on the address. The complete binary information was then converted to a digital file. The data recovery rate was calculated using $\frac{\text{successfully recovered binary segments}}{\text{total number of binary segments}}$ (Supplementary Data 1). For error analysis, sequencing data with average depth of 100×, 300×, 500×, 700× and 900× were randomly subsampled six times using different random seeds.

In total, >50 M PE-100 reads were generated for *in vivo* storage, in which the 10% low-quality reads (Phred score <20) by SOAPnuke were filtered[49]. Reads of the host genome were removed using samtools after mapping by BWA[50,51]. Short reads were then assembled into contigs by SOAPdenovo[52,53]. Blastn was used to find the connections between contigs[54]. A Python script was written to merge the contigs and obtain the assembled sequences for each strain. Multiple sequence alignment was conducted to align the assembled sequences by clustalW2 for the majority voting process to identify structural variations, insertions and deletions[55]. Pre-added RS codes were used for error correction of substitutions. The complete DNA sequence was decoded by reversing the operations of encoding to recover the binary information.

The physical density was calculated as

$$\frac{\text{Average information carried per nucleotide}}{\text{Average mass per nucleotide} \times \text{Average copy number} \times (1+\text{Redundancy percentage})},$$

where

$$\text{Average mass per nucleotide} = \frac{\text{Average molecular weight per nucleotide}}{\text{Avogadro constant}}$$

and

$$\text{Average information carried per nucleotide}$$
$$= \frac{2 \times \text{Number of nucleotides in data payload region}}{\text{Total length used}}.$$

The average molecular weight per nucleotide is 330.95 g mol$^{-1}$, which is a constant.

For the *in vitro* demonstration in this work, the average copy number for effective data recovery is 100, the length of the data payload region is 128 nt, the total length is 200 nt and the redundancy is ~30% including the 'pseudobinary' sequence. Therefore, the physical density is calculated to be $1.79 \times 10^{19}$ bits per gram of DNA, which equals $2.25 \times 10^{18}$ bytes per gram of DNA.

For the *in vivo* demonstration in this work, the average copy number is 1 as the exogenous sequence is integrated into genome, the length of the data payload region is 51,528 bp and the total length is 54,240 bp. Therefore, the physical density is calculated to be $3.46 \times 10^{21}$ bits per gram of DNA, which equals $4.322 \times 10^{20}$ bytes per gram of DNA.

For Chen et al., according to their paper[30], the average copy number is 1, the average information carried per nucleotide is 1.19 bits nt$^{-1}$. Therefore, the physical density is calculated to be $2.707 \times 10^{20}$ bytes per gram of DNA.

**Reporting Summary.** Further information on research design is available in the Nature Research Reporting Summary linked to this article.

## Data availability

Source data for all figures are provided with this paper. The sequencing raw data that support the findings of this study have been deposited in the CNSA (https://db.cngb.org/cnsa/) of the CNGBdb with accession code CNP0001650.

## Code availability

The code package for the YYC is available in the GitHub repository (https://github.com/ntpz870817/DNA-storage-YYC) and Zenodo[56].

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

## Acknowledgements

This work was supported by the National Key Research and Development Program of China (no. 2021YFF1200100, no. 2020YFA0712100), National Natural Science Foundation of China (no. 32101182) and Guangdong Provincial Key Laboratory of Genome Read and Write (no. 2017B030301011). We thank C. Hunter and C.-T. Wu from Harvard University and G. Ge from Capital Normal University for constructive discussions on the theoretical modelling. We thank the China National GeneBank (CNGB) for support with DNA fragment synthesis and assembly for the *in vivo* storage experiment.

## Author contributions

Y.S., Z.P., S.C., G.Z. and X.H. designed the experiment. Z.P. and S.C. conducted simulation and data analysis. G.Z. conducted the sequencing data analysis. S.J.Z. and H.Z. wrote and improved the code of the software program. J.C. and T.C. conducted the *in vivo* DNA fragment assembly. Z.L., H.Z. and Z.P. conducted the theoretical justification. Z.P. and Y.S. drafted the manuscript. Z.P., H.Z. and Y.S. prepared the figures and tables. G.Z., S.J.Z., H.H.L., G.M.C. and Y.S. revised the manuscript. H.Y., X.X., G.M.C. and Y.S. supervised the study. All authors read and approved the final manuscript.

## Competing interests

S.Z. is currently the founder of TAICHI AI Ltd, 20-22, Wenlock Road, London, England, N1 7GU. This work was completed when S.Z. was working at the University of Oxford and consulting for the BGI. G.M.C. has significant interests in Twist, Roswell, BGI, v.ht/PHNc, and v.ht/moVD. X.H., S.C., T.C., Y.S., X.X., and H.Y. have a patent filed with application number 16/858,295 and publication number 20200321079. The remaining authors declare no competing interests.

## Additional information

**Correspondence and requests for materials** should be addressed to Xun Xu, George M. Church or Yue Shen.

George M. Church
Yue Shen

# Reporting Summary

Nature Research wishes to improve the reproducibility of the work that we publish. This form provides structure for consistency and transparency in reporting. For further information on Nature Research policies, see our Editorial Policies and the Editorial Policy Checklist.

## Statistics

For all statistical analyses, confirm that the following items are present in the figure legend, table legend, main text, or Methods section.

| n/a | Confirmed | |
|---|---|---|
| ☐ | ☒ | The exact sample size (*n*) for each experimental group/condition, given as a discrete number and unit of measurement |
| ☐ | ☒ | A statement on whether measurements were taken from distinct samples or whether the same sample was measured repeatedly |
| ☐ | ☒ | The statistical test(s) used AND whether they are one- or two-sided <br> *Only common tests should be described solely by name; describe more complex techniques in the Methods section.* |
| ☐ | ☒ | A description of all covariates tested |
| ☐ | ☒ | A description of any assumptions or corrections, such as tests of normality and adjustment for multiple comparisons |
| ☐ | ☒ | A full description of the statistical parameters including central tendency (e.g. means) or other basic estimates (e.g. regression coefficient) AND variation (e.g. standard deviation) or associated estimates of uncertainty (e.g. confidence intervals) |
| ☒ | ☐ | For null hypothesis testing, the test statistic (e.g. *F*, *t*, *r*) with confidence intervals, effect sizes, degrees of freedom and *P* value noted <br> *Give P values as exact values whenever suitable.* |
| ☒ | ☐ | For Bayesian analysis, information on the choice of priors and Markov chain Monte Carlo settings |
| ☒ | ☐ | For hierarchical and complex designs, identification of the appropriate level for tests and full reporting of outcomes |
| ☐ | ☒ | Estimates of effect sizes (e.g. Cohen's *d*, Pearson's *r*), indicating how they were calculated |

*Our web collection on statistics for biologists contains articles on many of the points above.*

## Software and code

Policy information about availability of computer code

| Data collection | We published our custom python codes on Github and Zenodo. Source code and user manual are also available under: https://github.com/ntpz870817/DNA-storage-YYC and https://doi.org/10.5281/zenodo.6326563. |
|---|---|
| Data analysis | All encoding, decoding, and error analyzing experiments were performed in an Ubuntu 16.04.7 environment including an i7 CPU and 16 GB of RAM using Python 3.7.3, with our developed package "Chamaeleo" available under: https://github.com/ntpz870817/Chamaeleo. <br> For NGS result analysis, we used BWA v0.7.13, samtools v0.1.19-44428cd and vSOAP 2.7.7. For multiple sequence alignment, we used clustalW2 (CLUSTAL v2.1). |

For manuscripts utilizing custom algorithms or software that are central to the research but not yet described in published literature, software must be made available to editors and reviewers. We strongly encourage code deposition in a community repository (e.g. GitHub). See the Nature Research guidelines for submitting code & software for further information.

## Data

Policy information about availability of data

All manuscripts must include a data availability statement. This statement should provide the following information, where applicable:
- Accession codes, unique identifiers, or web links for publicly available datasets
- A list of figures that have associated raw data
- A description of any restrictions on data availability

The data that support the findings of this study have been deposited in the CNSA (https://db.cngb.org/cnsa/ ) of CNGBdb with accession code CNP0001650.
The figures-associated raw data is provided with the paper.
No restriction is on data availability.

# Field-specific reporting

Please select the one below that is the best fit for your research. If you are not sure, read the appropriate sections before making your selection.

☒ Life sciences          ☐ Behavioural & social sciences          ☐ Ecological, evolutionary & environmental sciences

For a reference copy of the document with all sections, see nature.com/documents/nr-reporting-summary-flat.pdf

# Life sciences study design

All studies must disclose on these points even when the disclosure is negative.

| | |
|---|---|
| Sample size | 'Sample size' was determined by technical properties of DNA synthesis and sequencing. Different kinds and formats of data were chosen for DNA-based data storage, which is sufficient to prove the universality and robustness of this codec. |
| Data exclusions | Raw sequencing data was filtered under a pre-established criteria, 10% low-quality reads (Phred score < 20) by SOAPnuke were filtered. |
| Replication | The source data retrieval experiments were repeated twice for two batches of synthesized oligo pools. The attempts were always successful for the replications. The original data was successfully decoded in all technical repeats, or replications.  The experiments of second batch were conducted about one year after the those of first batch. |
| Randomization | The simulations were run with precisely defined parameter value settings. Experimental validation also used precisely defined sequences and standard operation. Thus, randomization is not relevant to the study. |
| Blinding | The simulations were run with precisely defined parameter value settings. Experimental validation also used precisely defined sequences and standard operation. Thus, blinding is not relevant to the study. |

# Reporting for specific materials, systems and methods

We require information from authors about some types of materials, experimental systems and methods used in many studies. Here, indicate whether each material, system or method listed is relevant to your study. If you are not sure if a list item applies to your research, read the appropriate section before selecting a response.

## Materials & experimental systems

| n/a | Involved in the study |
|---|---|
| ☒ ☐ | Antibodies |
| ☒ ☐ | Eukaryotic cell lines |
| ☒ ☐ | Palaeontology and archaeology |
| ☒ ☐ | Animals and other organisms |
| ☒ ☐ | Human research participants |
| ☒ ☐ | Clinical data |
| ☒ ☐ | Dual use research of concern |

## Methods

| n/a | Involved in the study |
|---|---|
| ☒ ☐ | ChIP-seq |
| ☒ ☐ | Flow cytometry |
| ☒ ☐ | MRI-based neuroimaging |

