## [Peer Review File · Nature Computational Science]

Peer Review Information

Journal: Nature Computational Science

Manuscript Title: Towards Practical and Robust DNA-Based Data Archiving Using 'Yin-Yang Codec' System

Corresponding author name(s): Yue Shen, George M. Church, Xun Xu

Reviewer Comments & Decisions:

Decision Letter, initial version:

Date: 1st September 21 10:48:22

Last Sent: 1st September 21 10:48:22

Triggered By: Ananya Rastogi

From: ananya.rastogi@nature.com

To: shenyue@genomics.cn

BCC: ananya.rastogi@nature.com

Subject: Decision on Nature Computational Science manuscript NATCOMPUTSCI-21-0438

Message: ** Please ensure you delete the link to your author homepage in this e-mail if you wish to forward it to your co-authors. **

Dear Dr Shen,

Your manuscript "Towards Practical and Robust DNA-Based Data Archiving Using 'Yin-Yang Codec' System" has now been seen by 3 referees, whose comments are appended below. You will see that while they find your work of interest, they have raised points that need to be addressed before we can make a decision on publication.

The referees' reports seem to be quite clear. Naturally, we will need you to address all of the points raised.

While we ask you to address all of the points raised, the following points need to be substantially worked on:

- Please provide a more appropriate experiment to assess YYC's performance against DNA fountain as requested by Reviewer #2.
- Please provide an argument as to how any unique feature of YYC is relevant for in vivo storage besides the higher coding density.

- As pointed out by Reviewer #2, please discuss the impact of parameter choices made in the study.
- Please present the mathematical analysis in the study in a way which is accessible to a broad readership.

Please use the following link to submit your revised manuscript and a point-by-point response to the referees' comments (which should be in a separate document to any cover letter):

[REDACTED]

** This url links to your confidential homepage and associated information about manuscripts you may have submitted or be reviewing for us. If you wish to forward this e-mail to co-authors, please delete this link to your homepage first. **

To aid in the review process, we would appreciate it if you could also provide a copy of your manuscript files that indicates your revisions by making use of Track Changes or similar mark-up tools. Please also ensure that all correspondence is marked with your Nature Computational Science reference number in the subject line.

In addition, please make sure to upload a Word Document or LaTeX version of your text, to assist us in the editorial stage.

To improve transparency in authorship, we request that all authors identified as 'corresponding author' on published papers create and link their Open Researcher and Contributor Identifier (ORCID) with their account on the Manuscript Tracking System (MTS), prior to acceptance. ORCID helps the scientific community achieve unambiguous attribution of all scholarly contributions. You can create and link your ORCID from the home page of the MTS by clicking on 'Modify my Springer Nature account'. For more information please visit www.springernature.com/orcid.

We hope to receive your revised paper within three weeks. If you cannot send it within this time, please let us know.

Best regards,

Ananya Rastogi, PhD
Associate Editor
Nature Computational Science

Reviewers comments:

Reviewer #1 (Remarks to the Author):

Motivated by the Goldman's rotating encoding strategy and the DNA fountain encoding strategy, this paper presents an interesting encoding scheme called as Yin-Yang Codec

(YYC) for archival DNA data storage. The YYC scheme has three main steps. In the first step, the byte strings are partitioned into segments of equal length. In the second step of incorporation, two binary segments are selected randomly and are combined bit by bit using Yang rule first and then Yin rule, giving a final nucleotide as an output. In the final step, the generated DNA strings are screened for pre-set constraints for GC content (40%-60%), maximum homopolymer length < 5 , and the secondary structure free energy (≥ -30 kcal/mol). In a failed case, through an iteration process another randomly generated binary segment is used. It is observed that there are 1536 options for encoding the binary sequence. These different options give a maximum information density (theoretical) of 1.965 bits per base under specific constraints.

This encoding has been simulated on a computer and compared with well-known encoding schemes using a developed tool called Chamaeleo. The fountain DNA code and the YYC are specifically compared. The robustness of the YYC scheme was tested by introducing random and systematic errors and it was found that the data recovery percentage can be maintained at 98% when the sequence loss rate is $< 2\%$.

The experimental validation of the work was done using two file formats and these were stored in in vitro as 200 nts oligos and in vivo as an 54240-bp DNA fragment in yeast cell. This resulted in physical information density of ~ 198.8 Exabytes per gram of DNA. This is much better than the previous work in the literature.

I think the present work is an interesting approach. The introduced YYC scheme outperforms many previous schemes in terms of reliable data storage and physical information density. Some of my observations are listed below:

1. I feel that the YYC encoding scheme can be better presented. There are a few problems in the current writing. For example, in Figure 1, the choice of alphabet for Xn and Yn should be given. Although a working example is given in Figure S1, the presentation is from bits and nucleotides to nucleotides whereas in the general scheme it is nucleotides and nucleotides to bits. Another working example may be given either in the supplementary files or in the paper itself that will make it very clear.
2. I could not find the supplementary video mentioned at page 4 and page 14. Perhaps, including that will also be helpful.
3. Also, in the YYC scheme, there are 1536 possible options, so maybe it would be interesting to include a summary of all the rules in a table. This table could also include the rule number such as rule no. 888 that is used in Figure S1 or other rules in supplementary information. It is not clear how these rule numbers are assigned? Do they have some significance? I mean why the specific rule 888 was used.
4. At page 5, it is mentioned that when most of the binary segment contains more than 80% repeated 0's and 1's, the corresponding encoding process may enter an infinite loop or generate a poor result. Can we give some algebraic explanation for this from the YYC scheme?
5. At page 12, statement "YYC offers the opportunity to incorporate multiple rules for one file transcoding and thus provide a novel strategy for secure data archiving." Can we give a specific example? This will motivate the reader.
6. At page 15, line 10, What is F_q in Definition 1, line 1? Normally, F_q is a finite field having q elements but here, F_q is a set of 4 nucleotides, so q has no role to play. Maybe, the notation can be modified or F_q can be defined in line 1 and later, we can choose it to be an alphabet of size 4.

With these suggestions, I recommend for the publication of the manuscript.

Manish K Gupta

Reviewer #2 (Remarks to the Author):

The authors have introduced an interesting DNA encoding scheme called YYC (or Ying-Yang coding) with complementary rule sets to increase the coding density of current encoding methods. It is a dynamic transcoding scheme that transforms binary data into DNA sequences that could get close to the theoretical maximum information sensitivity of 2 bits per base. While the encoding approach is novel, there are inconsistencies and lack of exploration of parameters in the manuscript that make it difficult (or impossible) for anyone to assess the claimed merits of the work. It requires massive revisions, proper parameter studies, and significant retooling of how this paper is compared or "advertised" against other works.

Summarized below are the major concerns:

1. Data recovery rate - in the manuscript, DNA oligos are present at roughly $1e8$ copies per designed oligo sequence, which by my calculation is ~ 175 nanograms of total synthesis mass for Pool 1. They then dilute this down $10,000x$ fold in order to show that YYC/DNA fountain can get good recovery, but it completely breaks down at further serial dilutions. This is somewhat of a contrived experiment because it means the first dilution (10^4 molecules, which by my calculation of $10^{103} * (10^4 \text{ molecules}) * (200 * 330\text{g/mol}) * (1 \text{ mol} / (6.02 * 10^{23} \text{ molecules}))$) is 10 picograms of DNA. That is already an incredibly tiny amount of DNA. Given the data presented, it's not clear whether YYC's benefits are marginal or substantial. A more appropriate experiment to assess YYC's performance against DNA fountain in this context would be to show its performance for higher input copies (eg. $1e7$, $1e6$, etc.), as well as performances of re-amplifying pools from smaller dilutions (eg. dilution down to $1e7$, $1e6$, $1e5$, amplifying back to $1e8$ and using that as input).

2. In vivo storage - This part actually has very little to do with the YYC encoding and I don't understand its relevance to the manuscript's presentation of YYC encoding as a whole. You assemble DNA oligonucleotides or amplicons of any kind and clone into a plasmid for storage. The authors did not make any kind of argument about how any unique feature of YYC is relevant for in vivo storage besides the higher coding density, as you could theoretically just use any other high performance code. Interestingly, the authors did not touch on this point at all.

Furthermore, one of the most eye-catching errors in the paper is the advertisement of the storage density of $\sim 200\text{EB/g}$. Storage densities, as reflected in bits of information per mass, is limited by the synthesis of unique fragments and also by the coding density of the encoding procedure. Given these considerations, it is impossible that YYC achieves densities of $\sim 1000x$ higher than DNA fountain (and other works in Table 1) despite similar orders of magnitude in coding density. Even looking at the authors' formula as to how to calculate their vivo storage metric, it appears that nothing in that formula is fundamentally unique to the YYC coding scheme or that by plugging in the stated coding density of DNA fountain would get you a $\sim 1000x$ difference in performance. In fact, the differences are marginal or negligible when plugging in

different coding densities (into the information/nucleotide section, adjusting for plasmid backbone size) from Table 1.

Investigating the calculation further (by briefly reading the DNA fountain work which this manuscript cites often), it appears that there is a fundamental difference in how the calculations were determined. It appears that Erlich et al cited their 215PB metric as part of a serial dilution scheme for assessing successful/perfect retrieval of synthesized DNA. Taking the logic from the Erlich paper, the authors should have diluted down their yeast preps accordingly as well (to specific copies of plasmid loaded per sequencing library). Further, there seems to be a methods typo, where the authors stated that they picked three colonies but later in the methods it says they picked 12.

3. Other concerns regarding YYC encoding - it's clear that there is some sort of performance dependence according to the 1536 transcoding rules (Fig. S3 shows this pretty well). Across the manuscript there are tables with selected parameters but little discussion of the exact impact of them. It's difficult to assess what exactly the impact this has on performance when these parameters were not exhaustively studied either in silico or through actual experiments. Given the fact that this is a new coding scheme, the authors should have at minimum done an in silico demonstration of every single rule and how it affects every single metric in the paper. Off the top of my head, because the selection of the rule parameter greatly affects the information density, sequencing errors will play a large role in the overall performance.

Reviewer #3 (Remarks to the Author):

The paper presents a new efficient coding scheme that improves the recovery rate in case of large sequence loss. The method maintains linear dependency between the percentage of oligos that were fully recovered, and the data recovery. Additionally, the paper demonstrates in vivo data storage with high information density. Roughly speaking, the paper is fairly written, however there are several parts which are not well explained, as will be elaborated below.

Comments:

1. The code works well, I used it through the "chameleo" tool (that was also published by this group).
2. Like I sent you in another email - I could not access the data from their experiment, so either the link is broken, or they should add more explanations on the data and how to access it.
3. As far as I understand, the authors did not perform any reconstruction algorithm on the sequenced oligos. Can you please comment whether such an algorithm can improve the data recovery success and/or achieve faster data recovery?
4. There are several terms in the introduction that should be better explained before used for the average reader. Examples of these expressions are: secondary structure, free energy calculation.
5. The sentence between lines 80-82 is not clear to me.
6. Description of the YYC between lines 114-126 is not clear. What do X_n and Y_n refer to?
7. The caption of Fig. 1 is not clear either.
8. The description in page 11 of lines 282-310 is not clear to me.
9. I could not find the supplementary video of Fig. S1b.

10. The statement in Lemma 3 is not properly stated.
11. In general, all the mathematical analysis in page 15 is very sloppy. The equations are explained and their correctness is not justified. There should be a clear statement about the properties that the YCC scheme satisfies and a proof for the redundancy. As written this way, I cannot truly understand and evaluate the code construction.
12. The authors compare mostly with fountain codes. What about comparison with the other coding schemes?

Author Rebuttal to Initial comments

Reviewers comments:

Reviewer #1 (Remarks to the Author):

Motivated by the Goldman's rotating encoding strategy and the DNA fountain encoding strategy, this paper presents an interesting encoding scheme called as Yin-Yang Codec (YYC) for archival DNA data storage. The YYC scheme has three main steps. In the first step, the byte strings are partitioned into segments of equal length. In the second step of incorporation, two binary segments are selected randomly and are combined bit by bit using Yang rule first and then Yin rule, giving a final nucleotide as an output. In the final step, the generated DNA strings are screened for pre-set constraints for GC content (40%-60%), maximum homopolymer length < 5 , and the secondary structure free energy (≥ -30 kcal/mol). In a failed case, through an iteration process another randomly generated binary segment is used. It is observed that there are 1536 options for encoding the binary sequence. These different options give a maximum information density (theoretical) of 1.965 bits per base under specific constraints.

This encoding has been simulated on a computer and compared with well-known encoding schemes using a developed tool called Chamaeleo. The fountain DNA code and the YYC are specifically compared. The robustness of the YYC scheme was tested by introducing random and systematic errors and it was found that the data recovery percentage can be maintained at 98% when the sequence loss rate is $< 2\%$.

The experimental validation of the work was done using two file formats and these were stored in in vitro as 200 nts oligos and in vivo as an 54240-bp DNA fragment in yeast cell. This resulted in physical information density of ~ 198.8 Exabytes per gram of DNA. This is much better than the previous work in the literature.

I think the present work is an interesting approach. The introduced YYC scheme outperforms many previous schemes in terms of reliable data storage and physical information density. Some of my observations are listed below:

1. I feel that the YYC encoding scheme can be better presented. There are a few problems in the current writing. For example, in Figure 1, the choice of alphabet for X_n and Y_n should be given. Although a working example is given in Figure S1, the presentation is from bits and nucleotides to nucleotides whereas in the general scheme it is nucleotides and nucleotides to bits. Another working example may be given either in the supplementary files or in the paper itself that will make it very clear.

Response 1: We thank the reviewer for the comment. We have revised the manuscript accordingly to include detailed description in the main text highlighted on the page 4 (line 121-132) and figure S1 in supplementary. In addition, we have also re-uploaded a step-by-step demonstration video to show the encoding process for further elaboration.

2. I could not find the supplementary video mentioned at page 4 and page 14. Perhaps, including that will also be helpful.

Response 2: We apology to the reviewer on this. We have re-uploaded the video file as “supplementary video S1” in the supplementary for further description.

3. Also, in the YYC scheme, there are 1536 possible options, so maybe it would be interesting to include a summary of all the rules in a table. This table could also include the rule number such as rule no. 888 that is used in Figure S1 or other rules in supplementary information. It is not clear how these rule numbers are assigned? Do they have some significance? I mean why the specific rule 888 was used.

Response 3: We thank the reviewer for the comment. The rule number is assigned in numeric order from Rule no. 1 to Rule no. 1536. We followed the reviewer’s suggestion to include a summary of all rules in the main text on page 6 (line 172 - 180) and Figure S2 to demonstrate the general performance regarding information density of all 1,536 coding schemes using 1 GB testing data including text, image, audio and video files. As screening criteria settings on sequence compatibility, the GC content, max homopolymer all fell into the criteria of constraints. The corresponding results are described in the main text highlighted on page 6 (line 183 - 198) and Table

1. In this study, No. 888 was selected as a representing coding scheme with information density at medium level.

4. At page 5, it is mentioned that when most of the binary segment contains more than 80% repeated 0's and 1's, the corresponding encoding process may enter an infinite loop or generate a poor result. Can we give some algebraic explanation for this from the YYC scheme?

Response 4: We thank the reviewer for the comment. We followed the reviewer's suggestion to include an analysis to show the general performance of 1,536 combinatoric rules. The analysis is based on incorporation of two short binary segments (each binary segment contains 8 bits, which give 65,536 different combinations) based on its ratio of binary digit 0 and 1. When 0/1 ratio is less than 20%, it is shown that the number of coding schemes that can generate valid DNA sequence drops significantly from 100% to 49.7% and even lower when 0/1 ratio keeps decreasing. Therefore, 0/1 biased binary segments are firstly separated to improve the encoding efficiency. We set a "firewall" to limit the iteration run time at 100. For extreme cases, when most of the binary segments are 0/1 biased, "pseudo" binary segments with random 0/1 but in balanced ratio will be introduced to allow the generation of valid sequence. We have updated with description in the main text highlighted on page 5 (line 148 - 162) and Table S2 for further declaration.

5. At page 12, statement "YYC offers the opportunity to incorporate multiple rules for one file transcoding and thus provide a novel strategy for secure data archiving." Can we give a specific example? This will motivate the reader.

Response 5: We thank the reviewer for the comment. Generally, for other developed methods, the coding process follows a unitary transcoding rule. In contrast, YYC offers in total 1536 combinatorial transcoding rules, that each rule will generate distinct DNA sequencing. Thus, YYC offers the possibility to use multiple rules in one file transcoding and therefore could significantly increase the difficulty of deciphering the coding rule by brute-force attacks or other manners. We have revised this in the main text highlighted on page 14 (line 384 – 387).

6. At page 15, line 10, What is F_q in Definition 1, line 1? Normally, F_q is a finite field having q elements but here, F_q is a set of 4 nucleotides, so q has no role to play. Maybe, the notation can be modified or F_q can be defined in line 1 and later, we can choose it to be an alphabet of size 4.

Response 6: We thank the reviewer for the comment. We have revised this according to the reviewer's suggestion. The choice of two binary digits and four nucleotides is stated in "Quantitative analysis" section of the supplementary. We have rewritten this part to make it easier to understand.

With these suggestions, I recommend for the publication of the manuscript.

Manish K Gupta

Response 7: We thank the reviewer for the recognition on YYC and significant help on improving our work. We hope our revised manuscript will reach to the level of publication on *Nature Computational Science*.

Reviewer #2 (Remarks to the Author):

The authors have introduced an interesting DNA encoding scheme called YYC (or Ying-Yang coding) with complementary rule sets to increase the coding density of current encoding methods. It is a dynamic transcoding scheme that transforms binary data into DNA sequences that could get close to the theoretical maximum information sensitivity of 2 bits per base. While the encoding approach is novel, there are inconsistencies and lack of exploration of parameters in the manuscript that make it difficult (or impossible) for anyone to assess the claimed merits of the work. It requires massive revisions, proper parameter studies, and significant retooling of how this paper is compared or "advertised" against other works.

Summarized below are the major concerns:

1. Data recovery rate - in the manuscript, DNA oligos are present at roughly $1e8$ copies per designed oligo sequence, which by my calculation is ~ 175 nanograms of total synthesis mass for Pool 1. They then dilute this down $10,000x$ fold in order to show that YYC/DNA fountain can get good recovery, but it completely breaks down at further serial dilutions. This is somewhat of a contrived experiment because it means the first dilution (10^4 molecules, which by my calculation of $10103 * (10^4 \text{ molecules}) * (200 * 330\text{g/mol}) * (1 \text{ mol} / (6.02 * 10^{23} \text{ molecules}))$) is 10 picograms of DNA. That is already an incredibly tiny amount of DNA. Given the data presented, it's not clear whether YYC's benefits are marginal or substantial. A more appropriate experiment to assess YYC's performance against DNA fountain in this context would be to show its performance for higher input copies (eg. $1e7$, $1e6$, etc.), as well as performances of re-amplifying pools from smaller dilutions (eg. dilution down to $1e7$, $1e6$, $1e5$, amplifying back to $1e8$ and using that as input).

Response 8: We thank the reviewer for the comment. As for the reason to perform dilution assay at the starting level of $1e4$ copies: The three master pools of 200nt DNA oligos we ordered from TWIST Bioscience are delivered in the form of lyophilized powder (190 ng, 182 ng and 172 ng respectively for P1, P2 and P3) with the yield of ~ 0.2 - 1 fmol per oligo, which equals to the copy number at $\sim 1e8$ as indicated by the reviewer. The reason to perform dilution assay at the starting level of $1e4$ copies is that in Erlich et. al, 2017 of DNA fountain, they validated the successful recovery of stored information at the level of $1e3$ copies. Then in our study, we used the same setting as claimed in the paper and proven that both YYC and DNA fountain can perfectly recover stored data at the level of $1e4$ copies, indicating that there will be no difference between the two methods on the recovery rate for higher input at $1e5$, $1e6$ and $1e7$ copies.

To perform data recovery rate assessment under low input copies, given that data retrieval from master pool is a consuming process for synthesized DNA molecules and synthesizing substantial DNA is still too costly at the moment, thus in our study, further serial dilutions from $1e3$ down to the $1e1$ is designed to evaluate the data recovery performance of YYC under low amount of synthesized DNA.

However, we revised the experiment design according to the reviewer's suggestion by including the experimental evaluation from high input (at $1e7$ copies in this study) all the way to low input (at $1e0$ copies in this study). The DNA samples were re-ordered from the same vendor (TWIST Bioscience) we used in this study to repeat the whole experiment. In general, our updated results show that YYC exhibits relatively better performance regarding data recovery. With the input of $1e4$ to $1e6$ copies, the data recovery rate of YYC remains stable at comparable level with DNA fountain. And for the input of low copies ($1e2$ to $1e3$ copies), YYC shows superior performance in data recovery. In addition, we found that at extreme low input level ($1e0$ to $1e1$ copies), even the DNA molecules were re-amplified to $1e8$ copies, the data recovery rate still cannot be guaranteed.

We have updated all related results shown in main text highlighted on page 10-12 (line 267 - 325), Figure 3, methods "*Library preparation and sequencing*" section on page 19 (line 543 - 558), supplementary excel file, Sheet 2 & 3 accordingly.

2. In vivo storage - This part actually has very little to do with the YYC encoding and I don't understand its relevance to the manuscript's presentation of YYC encoding as a whole. You assemble DNA oligonucleotides or amplicons of any kind and clone into a plasmid for storage. The authors did not make any kind of argument about how any unique feature of YYC is relevant for in vivo storage besides the higher coding density, as you could theoretically just use any other high performance code. Interestingly, the authors did not touch on this point at all.

Response 9: We thank the reviewer for the comment. The fast technical development of DNA synthesis and assembly promotes both *in vitro* and *in vivo* storage as well recognized approaches in recently years (Shipman et. al, *Nature*, 2017, Tabatabaei et. al, *Nature Communications*, 2020 and Chen et. al, *National Science Review*, 2021). Thus, we include both approaches in our study for experimental demonstrations. Previous study suggest that in vivo storage holds the advantage of retrieving data can be easily performed by cell subculturing and DNA extraction, and stored data can be well maintained. However, we noticed that introducing substantial heterogenous DNA into host strains could be problematic for data recovery. By sequencing of multiple single colonies isolated from the plate, we observed partial data coding DNA loss in

varying degrees. Our result further suggests that data recovery performance of the coding scheme is essentially important. It is common to observe spontaneous mutations or unexpected variations accumulated during long term passaging of living cells. Therefore, it would be very important to apply more robust coding algorithms to retain as much original information as possible. We thank the reviewer for pointing out the importance of providing adequate justification on the reason of doing *in vivo* storage. We have revised the main text highlighted on page 12 (line 328 - 355), Figure 4, methods “*Synthesis and assembly*” section on page 19 (line 537 - 540), methods “*Data analysis*” section on page 20 (line 575 - 601) and supplementary Table S7 to further clarify the importance of robustness of coding algorithm for *in vivo* storage.

Furthermore, one of the most eye-catching errors in the paper is the advertisement of the storage density of $\sim 200\text{EB/g}$. Storage densities, as reflected in bits of information per mass, is limited by the synthesis of unique fragments and also by the coding density of the encoding procedure. Given these considerations, it is impossible that YYC achieves densities of $\sim 1000\text{x}$ higher than DNA fountain (and other works in Table 1) despite similar orders of magnitude in coding density. Even looking at the authors' formula as to how to calculate their *vivo* storage metric, it appears that nothing in that formula is fundamentally unique to the YYC coding scheme or that by plugging in the stated coding density of DNA fountain would get you a $\sim 1000\text{x}$ difference in performance. In fact, the differences are marginal or negligible when plugging in different coding densities (into the information/nucleotide section, adjusting for plasmid backbone size) from Table 1.

Investigating the calculation further (by briefly reading the DNA fountain work which this manuscript cites often), it appears that there is a fundamental difference in how the calculations were determined. It appears that Erlich et al cited their 215PB metric as part of a serial dilution scheme for assessing successful/perfect retrieval of synthesized DNA. Taking the logic from the Erlich paper, the authors should have diluted down their yeast preps accordingly as well (to specific copies of plasmid loaded per sequencing library). Further, there seems to be a methods typo, where the authors stated that they picked three colonies but later in the methods it says they picked 12.

Response 10: We thank the reviewer for the comment. First, we would like to point out that information density is different concept from the physical density as we suggested in the manuscript on page 3 (line 83 - 87). Our argument is also supported previously by *Organick et. al., 2020*. Information density is determined by the algorithm itself, while the physical density indeed will vary by the selected approaches of storage (i.e. *in vitro* as DNA powder or *in vivo* as plasmid or integrated in the host genome). In the main text, we clarified that the physical density achieved in this study is based on *in vivo* storage on page 20 (line 584 - 601). And the calculation logic is consistent with previous study performed experimental demonstration using the approach of *in vivo* storage (Tabatabaei et. al, *Nature Communications*, 2020 and Chen et. al, *National Science Review*, 2021), which is not covered in Erlich et al.'s work.

We accept the reviewer's suggestion and have revised Table 1 by: 1) providing corresponding physical density of both *in vitro* and *in vivo* storage performed in our study and 2) to include the physical density achieved in the recent published *in vivo* storage study (Chen et. al, *National Science Review*, 2021) to avoid misleading conclusion for readers. The calculation formula is also provided in the method "Data analysis" section highlighted on page 20 (line 584 - 601) for further clarification. For the method typo, we apology for the unintentional mistake, we have revised the description in the main text highlighted on page 12 (line 340 -342), Figure 4 and method description "Synthesis and assembly" section on page 18 (line 533 - 540) to clarify the correct numbers.

3. Other concerns regarding YYC encoding - it's clear that there is some sort of performance dependence according to the 1536 transcoding rules (Fig. S3 shows this pretty well). Across the manuscript there are tables with selected parameters but little discussion of the exact impact of them. It's difficult to assess what exactly the impact this has on performance when these parameters were not exhaustively studied either in silico or through actual experiments. Given the fact that this is a new coding scheme, the authors should have at minimum done an in silico demonstration of every single rule and how it affects every single metric in the paper. Off the top of my head, because the selection of the rule parameter greatly affects the information density, sequencing errors will play a large role in the overall performance.

Response 11: We thank the reviewer for the comment. Parameters in Table 1 and the experimental design (Figure 3) is selected to be consistent with previous studies for the purpose of comparability.

We accept the reviewer's suggestion and have revised accordingly in the main text highlighted on page 6 (line 172-180) and page 8 (line 220 – 232) to provide the in-silico demonstration to systematically evaluate their performance of all 1536 coding schemes using 1 GB data collection including different formats of files (text, image, audio, and video). To conclude in general, we have shown that: 1) although the information density of each coding schemes in the 1536 collections varies, but in the range from 1.75 bits/nt to 1.78 bits/nt under well-accepted constraints. 2) the overall performance including InDels and SNVs is not affected by sequencing/synthesis errors.

Reviewer #3 (Remarks to the Author):

The paper presents a new efficient coding scheme that improves the recovery rate in case of large sequence loss. The method maintains linear dependency between the percentage of oligos that were fully recovered, and the data recovery. Additionally, the paper demonstrates in vivo data storage with high information density. Roughly speaking, the paper is fairly written, however there are several parts which are not well explained, as will be elaborated below.

Comments:

1. The code works well, I used it through the "chameleo" tool (that was also published by this group).

Response 12: We thank the reviewer for the recognition of YYC code and careful testing efforts made to evaluate on YYC.

2. Like I sent you in another email - I could not access the data from their experiment, so either the link is broken, or they should add more explanations on the data and how to access it.

Response 13: We apologize to the reviewer for that the access link was not working. We didn't notice that there were some technical problems on linking our raw data to the project site. We have solved the problem and now the reviewer should be able to access the raw data by:

<https://db.cngb.org/search/project/CNP0001650/>

The raw sequencing data can also be accessed from:

<https://ftp.cngb.org/pub/CNSA/data3/CNP0001650>

3. As far as I understand, the authors did not perform any reconstruction algorithm on the sequenced oligos. Can you please comment whether such an algorithm can improve the data recovery success and/or achieve faster data recovery?

Response 14: We thank the reviewer for the comment. For both *in vitro* and *in vivo* experimental validations performed in our study, we used the standard de novo assembly algorithm SOAPdenovo (Luo *et. al.*, 2012, Li *et. al.*, 2010) applied in the next-generation sequencing analysis workflow. According to our result, we found that data recovery success/improvement is not achieved solely by the reconstruction algorithm itself but determined by the coding scheme and affected by errors introduced during synthesis/sequencing. However, we do think there could be a better algorithm or strategy designed specifically for this application in the future would improve the transcoding efficiency as well as the data recovery rate by better identifying and eliminating errors. We have revised this according to the reviewer's suggestion in the main text highlighted on page 12 (line 328 – 355) and included corresponding citations for algorithms used for sequence reconstruction in this study in the "Data analysis" section of method highlighted on page 20 (line 575 - 583).

4. There are several terms in the introduction that should be better explained before used for the average reader. Examples of these expressions are: secondary structure, free energy calculation.

Response 15: We thank the reviewer for the comment. We have revised this according to the reviewer's suggestion in the main text highlighted on page 2 (line 63 – 67).

5. The sentence between lines 80-82 is not clear to me.

Response 16: We thank the reviewer for the comment. In previous studies, two types of “redundancy” have been proposed and applied in DNA storage (one at transcoding level, while the other one refers to biochemical operation level). The logical redundancy of different transcoding methods varies, for example, it is mandatory for DNA fountain but not necessary for YYC. However, the physical redundancy (using extra molecule copies by synthesis) can be applied to all developed transcoding methods. They both can play a positive role in data recovery, but functioning in fundamentally different approaches. Thus we want to distinguish the difference between the two types of redundancy. We have revised this accordingly to make it clear in the main text highlighted on page 3 (line 83 – 87) and added relevant references (Ref No. 23,24).

6. Description of the YYC between lines 114-126 is not clear. What do X_n and Y_n refer to?

Response 17: We thank the reviewer for the comment. Here we use $N1/N2/N3/N4$ to represents one of the four nucleic acids A/T/C/G. In ‘Yang’ rule, 0 and 1 are mapped to two nucleotides: for example, if A or T represents 0, then the only choice for 1 is C or G. In ‘Yin’ rule, 0 and 1 are also mapped to two nucleotides, but in a different manner: X_j and Y_j represent different binary digits 0 and 1. When j is an integer chosen from 1 to 8, $X_j+Y_j=1$ and $X_j \times Y_j=0$ (i.e. eight independent sets of X and Y, with X_j/Y_j being either 1/0 or 0/1). Since $N1$ and $N2$, or $N3$ and $N4$ represent identical binary digits in ‘Yang’ rule, but represent different digit in ‘Yin’ rule, it will only give one and only one consensus nucleotide during encoding. We have revised this accordingly to make it clearer in the main text highlighted on page 4 (line 121 - 132) and updated additional illustration in the supplementary video S1 for further clarification.

7. The caption of Fig. 1 is not clear either.

Response 18: We thank the reviewer for the comment. We have revised the figure caption accordingly to make it clearer in the main text highlighted on page 4 (line 121 - 132).

8. The description in page 11 of lines 282-310 is not clear to me.

Response 19: We thank the reviewer for the comment. The fast technical development of DNA synthesis and assembly promotes both *in vitro* and *in vivo* storage as well recognized approaches in recently years (*Shipman et. al, Nature, 2017, Tabatabaei et. al, Nature Communications, 2020 and Chen et. al, National Science Review, 2021*). Thus, we include both approaches in our study for experimental demonstrations. Previous study suggest that *in vivo* storage holds the advantage of retrieving data can be easily performed by cell subculturing and DNA extraction, and stored data can be well maintained. However, we noticed that introducing substantial heterogenous DNA into host strains could be problematic for data recovery. By sequencing of single colonies isolated from the construct, we observed partial data coding DNA loss in varying degrees. Our result further suggests that data recovery performance of the coding scheme is essentially important. We have revised the description accordingly to make it clearer in the main text highlighted on page 12 (line 328 - 355).

9. I could not find the supplementary video of Fig. S1b.

Response 20: We apology to the reviewer on this. We have re-uploaded the video file as “supplementary video S1” in the supplementary for further description.

10. The statement in Lemma 3 is not properly stated.

Response 21: We thank the reviewer for the comment. We have rewritten this part to make it clearer to understand. The new section is in the “Quantitative analysis” section of Supplementary.

11. In general, all the mathematical analysis in page 15 is very sloppy. The equations are explained and their correctness is not justified. There should be a clear statement about the properties that the YCC scheme satisfies and a proof for the redundancy. As written this way, I cannot truly understand and evaluate the code construction.

Response 22: We thank the reviewer for the comment. We have rewritten this part to make it clearer to understand. The new section is in the “Quantitative analysis” section of Supplementary.

12. The authors compare mostly with fountain codes. What about comparison with the other coding schemes?

Response 23: We thank the reviewer for the comment. As we explained in our introduction part on page 3, all early efforts have paved the way of facilitating the fast development of DNA storage, with focus on biochemical compatibility and/or chasing the extreme of information density. The comparison with all developed coding scheme is performed by encoding 1GB data collection in our study for in silico simulation analysis. Relative result is described in the “General principle and features of the Yin-Yang codec” part on page 6 (line 183 - 198) and supplementary Table S1. Since DNA fountain has performed comprehensive analysis to show its superior performance in comparison with other above mentioned coding schemes (Yaniv Erlich and Dina Zielinski, Science 2017), thus in our study, we choose to focus the comparison between DNA fountain and YYC for experimental validation comparison.

Decision Letter, first revision:

Date: 17th January 22 04:48:08

Last Sent: 17th January 22 04:48:08

Triggered By: Ananya Rastogi

From: ananya.rastogi@nature.com

To: shenyue@genomics.cn

BCC: ananya.rastogi@nature.com

Subject: Decision on Nature Computational Science manuscript NATCOMPUTSCI-21-0438A

Message: ** Please ensure you delete the link to your author homepage in this e-mail if you wish to forward it to your co-authors. **

Dear Dr Shen,

Your manuscript "Towards Practical and Robust DNA-Based Data Archiving Using 'Yin-Yang Codec' System" has now been seen by the 3 referees that we had contacted earlier. In addition, we contacted 1 more reviewer for a more technical review. The reviewers' comments are appended below. You will see that while they find your work of interest, they have raised points that need to be addressed before we can make a decision on publication.

While we ask you to address all of the points raised, the following points need to be substantially worked on:

- Please provide more insight into the way indices are assigned to segments along with the effect this has on pool capacity.
- Please discuss the policy for what to do when a sequence fails to encode.
- Please provide a discussion on where the proposed codec fits in and how it could be used with other approaches.
- Please provide an insight into the observed poor performance of DNA Fountain.

Please use the following link to submit your revised manuscript and a point-by-point response to the referees' comments (which should be in a separate document to any cover letter):

[REDACTED]

** This url links to your confidential homepage and associated information about manuscripts you may have submitted or be reviewing for us. If you wish to forward this e-mail to co-authors, please delete this link to your homepage first. **

To aid in the review process, we would appreciate it if you could also provide a copy of your manuscript files that indicates your revisions by making use of Track Changes or similar mark-up tools. Please also ensure that all correspondence is marked with your Nature Computational Science reference number in the subject line.

In addition, please make sure to upload a Word Document or LaTeX version of your text, to assist us in the editorial stage.

To improve transparency in authorship, we request that all authors identified as 'corresponding author' on published papers create and link their Open Researcher and Contributor Identifier (ORCID) with their account on the Manuscript Tracking System (MTS), prior to acceptance. ORCID helps the scientific community achieve unambiguous attribution of all scholarly contributions. You can create and link your ORCID from the home page of the MTS by clicking on 'Modify my Springer Nature account'. For more information please visit www.springernature.com/orcid.

We hope to receive your revised paper within three weeks. If you cannot send it within this time, please let us know.

Best regards,

Ananya Rastogi, PhD
Associate Editor
Nature Computational Science

Reviewers comments:

Reviewer #1 (Remarks to the Author):

The points you raised in the previous round of review have been satisfactorily addressed.
The paper can be accepted now.

Reviewer #2 (Remarks to the Author):

The authors have adequately addressed all of my concerns.

Reviewer #3 (Remarks to the Author):

The authors addressed most of my comments as well as the ones by the other two reviewers. I still have several concerns with the paper presentation and my previous comments (I was reviewer #3).

- In my third comments, I asked about the use of some reconstruction algorithm. From your explanation it is implied that a voting strategy was applied to generate a consensus sequence. What exactly does it mean and why don't you use other algorithms for reconstruction.
- In comment 12, I still don't see the rationale behind the comparison with only DNA fountain codes. In general, you should mathematically describe the problem that is solved in the paper and why Yin-Yang codes are better than other known schemes for this problem.

Reviewer #4 (Remarks to the Author):

Reviewer Summary

This work describes a novel codec for DNA storage called the Yin-Yang Codec (YYC). It's based on the observation that it is difficult to achieve both theoretical information densities and sequences that meet critical biomolecular constraints, such as avoiding homopolymers, imbalanced GC content, or harmful structures. The YYC is effective at reaching high density because it supports a variety of encoding strategies.

Here's a brief summary of how it works. A file is divided into segments of a fixed length as binary strings; an index is added to each segment to mark its location in the file. Then, two segments, say *a* and *b*, are selected at random and sent through a two-step encoding process (Yin and Yang). One bit is taken from each of the two segments and is encoded into a single nucleotide. Using 2 bits to encode each nucleotide is what gives it a high information density. The Yin step uses *a*[*i*] to select a set of possible encoding bases, which narrows it down to 2 of the 4 bases based on a look-up table, call this set *A*. The Yang step uses the *b*[*i*] bit and the previous nucleotide that was encoded to look up its encoding, which is also a set of possible nucleotides, let's call it set *B*. Then, the intersection of *A* and *B* is emitted as the output nucleotide. The Yin and Yang coding tables are designed to ensure a unique nucleotide is emitted. This process repeats for all bits in *a* and *b* until the sequence is generated. Because two bits are used to produce each base, it has the potential to reach the information theoretical limit of 2 bits / nt. However, after the sequence is created, it's sent through a filter to weed out bad sequences based on GC content, homopolymers, and thermodynamic analysis. If it fails this step, the sequence is thrown out. This process of selecting segments, encoding them, and checking them is repeated until enough good sequences are created to enable decoding of the file. No additional logical redundancy is included since this process is only meant as a way of encoding the data and is not meant to provide error correction.

Experiments are conducted *in silico*, *in vitro*, and *in vivo* to understand its utility and verify its usefulness for DNA storage. The YYC approach is applied to a suite of files of various formats and shown to successfully encode and decode while achieving good information densities. Its effectiveness is compared primarily to DNA Fountain given their overall similarity in goals and approach.

The most interesting part of the evaluation is in how well the codec works *in silico*. The *in vitro* studies are expected to work since they largely follow proven experimental methodologies. The *in vivo* studies add some new analysis to the field concerning storage and retrieval of data in yeast.

Significance

This work aims to further the field by presenting a new codec that provides high information density while still avoiding problematic sequences. This codec bears significant similarity to the goals of DNA Fountain, and that's why they compare against DNA Fountain. However, this work is not a Fountain code, and that enables some simplifications while also creating potential weaknesses.

Advantages:

+A potential advantage of this work is the straightforward encoding and decoding process with respect to segments and their indices. This work can embed the indices directly in the encoded segments and recover them through standard decoding, but fountain codes need to encode that information as separate meta-data or hard code a lookup table or enumerator function a priori in the encoder and decoder.

+Another potential advantage of this work is the large variety of Yin and Yang coding tables, up to 1536, that allow a wide variety of ways to encode data. However, this may be a weakness, as described below.

Disadvantages:

- A fountain code can pick an arbitrary number of segments to merge, but this can only pick 2. YYC makes up for this limitation to some degree by having non-unitary mapping rules, however, this is achieved using a trial and error process. The trial and error process is not described in the paper or in the supplementary material, as far as I could tell. While the YYC can allow up to 1536 different coding tables, it was not demonstrated in the work that all of these combinations are actually useful, and I suspect that many of the tables end up producing similar results and may not all be equally useful.

- Fountain codes and rateless codes, in general, can add an arbitrary number of additional symbols to the encoded file, but this work can pick from at most n^2 combinations of segments, where n is the number of segments in the file, giving it far fewer combinations to work with to find usable sequences than Fountain codes.

- This work shares the disadvantage of DNA Fountain of needing to verify a sequence after its encoded, which means that either the pair of segments or the mapping tables could have been a poor choice. This can lead to large encode-time overheads and many failed attempts when encoding strands. The number of trials used to encode data and its compute time is not reported, but it should be for a fair assessment of the work. The work does report a limit of 100 attempts per strand, but the total number of attempts while encoding a file is not reported.

- The policy for what to do when a sequence fails to encode is not described in much detail. I am concerned with the computational complexity of choosing a suitable table - how many attempts are needed? Also, I'm concerned about what happens if there is no suitable encoding for some part of the data. If 100 attempts are made and all fail, at this point, is a suitable synthetic sequence created just to ensure encodability? Is this synthetic sequence somehow always guaranteed to succeed or might multiple attempts be needed here as well? It would be good to quantify this as well.

Clarity and Context

* I think the work does not position itself as well as it could with respect to prior codecs or with respect to DNA Fountain. Many different codecs have been proposed to deal with various challenges of DNA storage. The presentation of where this one fits in and how it could be used with other approaches could be done with greater clarity. For example, this is really just part of the codec and doesn't include any new support for

error correction. Also, this article takes a fairly limited view of what is relevant and does not discuss deeply how other codes were designed and what their advantages and disadvantages are. Organick et al. [Nature Biotech 2018] deal with these same issues but in different ways. Also, DNA Fountain, and fountain codes in general, could be configured and tuned to work potentially much better than the DNA Fountain work that is compared against, but this is not done or considered. The claimed advantages of YYC over DNA Fountain may be true for the specific implementation of DNA Fountain that was compared against, but those advantages likely do not extend to an optimized Fountain code for data storage.

* The example of the Yin-Yang Codec given in the paper is hard to understand. I think you should move a similar example as in the video into the paper. Make it concrete so that readers can understand how one of the configurations actually works. You can always add more explanation on how to generalize it elsewhere. Without a good example, the paper is not self-contained and reviewers will need the supplementary material to make sense of the work, which is not ideal in my opinion.

- The way indices are assigned to segments is not explained in enough detail. Also, how segments are assigned to either the Yin or Yang rule is not explained. These are important details because they have a large bearing on the overall efficiency of the system. For example, suppose a file is partitioned into segments and each one is labeled with a unique binary index, of length k (2^{**k} total segments are possible). Now, suppose these segments are selected at random and passed arbitrarily into either the Yin or Yang rule. This means that k nucleotides are used to represent the index and 4^{**k} sequences are possible, but only 2^{**k} total indices are available in the file. This implies that indices have at most a coding density of 1 bit/base. Large files will need large indices, which substantially cuts into the claimed high density of the system since oligo synthesis has relatively hard limits on synthesized strand length. Other codecs which do not use a binary index will have a significant advantage in overall capacity and density. It's possible that I've misunderstood some aspect of how indices are assigned and used in this codec, so it would be good for this aspect of the paper to be improved.

Major Suggestions

1. The paper makes a strong claim about density that needs to be clarified. As mentioned earlier, the work does not explain well how indices are handled (as mentioned previously in my review), but the indices appear to achieve no better than a binary encoding (1 bit /base) density. One implication of this is that a high-density encoding is somewhat reliant on having small indices, which implies small file sizes and small capacity overall. This is an undesirable result for what should be a dense medium. Another implication of this is that the total capacity of the system (total unique strands) appears to be lower than those systems that achieve ternary indices, for example. While the claimed figures of merit for density are high, they hide the fact that the total capacity of a pool of DNA may be severely negatively impacted by binary indices. I would like to see a better explanation of how indices are handled and the effect this has on pool capacity.

2. The capabilities of the YYC codec with respect to the variety of mapping tables is not well justified. It's not clear that the ability to select one of the 1536 configurations

actually helps or not. The work claims to leverage a combinatoric scheme, but that is not demonstrated through an experiment, except to some degree in the small one that analyzes strands with large fractions of 0 or 1 to see if any mapping tables can encode them satisfactorily.

To really demonstrate the advantage of all these combinations, several things need to be shown. (1) That having a choice among mapping tables is helpful and leads to less overhead as compared to DNA Fountain. (2) That the choice of mapping table can be made efficiently. It would be interesting to report the number of failed attempts at selecting segments and encoding them. (3) That the overhead of holding the choice of mapping table in meta-data is a reasonable cost, even if done electronically. If a mapping table selection needs to be remembered per strand, then that implies approximately 11 bits of overhead per segment-pair, which for an exabyte scale system would add up to a very large cost. If it only needs to be remembered per file, it's a lower meta-data cost but higher compute cost to find a mapping. These trade-offs need some further explanations.

If this analysis is prohibitive to conduct, then I would suggest softening the claims over how useful these tables are.

3. It's surprising that DNA Fountain does so poorly in the in silico experiment (Figure 2). I agree with the general arguments that DNA Fountain will have high degree packets that include many segments. However, I'm not convinced this is the entire answer unless DNA Fountain mostly selects high degree packets. Also, some important information is missing in this analysis. It appears that no error correction is used. In that case, what is done with strands that have insertions, deletions, and substitutions? Without error correction support, such problems cannot even be detected much less corrected. So, are they treated as correct and allowed to pollute the file with errors? This raises questions about how the DNA Fountain approach is utilized. Is the DNA Fountain code base used for this analysis? If so, is it possible that something is going wrong in how it handles these errors? For example, since segments are more likely repeated in multiple high degree packets, it may be detecting errors and throwing them out, whereas YYC has no such detection capability. Instead, YYC may be making a best-effort attempt to keep the data. More details of how this analysis is conducted are needed to fully explain DNA Fountain's poorer behavior.

Minor Suggestions

1. Perhaps add a brief explanation for a few things in the thermodynamic screening, such as at what temperature was the simulated free energy screening conducted? And, what other parameters were used, if any? Why did you pick -30 kcal/mol as the cutoff?
2. In the text regarding in vitro experiments, please make it clear which encoding was used for each pool. I think P2 and P3 were done using DNA Fountain.
3. On Page 9, line 242-244, it says that loss of one strand in YYC can lead to the loss of two segments. This is true. But, isn't it possible that a segment is repeatedly selected for pairing due to some good pattern in its data, decreasing the odds it is lost? Or, do you prevent selecting the same sequence many times?

Thank you for an interesting paper.

Author Rebuttal, first revision:

Reviewers' comments:

Reviewer #1 (Remarks to the Author):

The points you raised in the previous round of review have been satisfactorily addressed.

The paper can be accepted now.

Response 1: We thank the reviewer for the significant help in improving our work.

Reviewer #2 (Remarks to the Author):

The authors have adequately addressed all of my concerns.

Response 2: We thank the reviewer for the significant help in improving our work.

Reviewer #3 (Remarks to the Author):

The authors addressed most of my comments as well as the ones by the other two reviewers. I still have several concerns with the paper presentation and my previous comments (I was reviewer #3).

- In my third comments, I asked about the use of some reconstruction algorithm. From your explanation it is implied that a voting strategy was applied to generate a consensus sequence. What exactly does it mean and why don't you use other algorithms for reconstruction.

Response 3: We thank the reviewer for the comment. The voting strategy here is the single-winner plurality voting strategy where each candidate sequence has the same weight. As this strategy is straightforward and effective with the stored data fully recovered, we chose this strategy over other algorithms to minimize complexity and avoid algorithmic bias. We have revised the coordinating manuscript on page 13 (line 376 to 378) to avoid confusion.

- In comment 12, I still don't see the rationale behind the comparison with only DNA fountain codes. In general, you should mathematically describe the problem that is solved in the paper and why Yin-Yang codes are better than other known schemes for this problem.

Response 4: We thank the reviewer for the comment. Excepting DNA fountain, early established coding schemes apply a single fixed rule for bit-to-base transcoding, in which no screening process is used. On the contrary, DNA Fountain and YYC are the only two known coding schemes that combine transcoding rules and screening as a whole process to make sure the generated DNA sequences can meet the biochemical constraints, such as GC content and maximum homopolymer length. Thus, we reason that direct mathematical analysis should be focused on the comparison between DNA Fountain and YYC due to the similarity of coding strategy between YYC and DNA Fountain. Both algorithms employ the strategy of incorporation of two or more binary segments and generate a corresponding DNA sequence/ information packet. The information packets generated by DNA fountain are topologically connected with each other and form a grid-like structure. For current information communication using cables or radio, this procedure is feasible because even if some packets are lost or with error, new packets can be immediately re-sent for successful data recovery. However, receiving information in the process of DNA storage is not synchronous. Thus, the errors or packets loss will lead to a domino effect or error propagation for DNA Fountain. In contrast, the information packets generated by YYC are mutually independent. The effect on data recovery caused by the errors or packets loss is greatly minimized for YYC.

In addition, as we explained in our previous response, we have included the performance comparison among all developed coding schemes by encoding 1GB data collection in our study for in silico simulation analysis (Table 1). And our result suggests that YYC shows superior performance in comparison with other existing coding schemes.

Reviewer #4 (Remarks to the Author):

Reviewer Summary

This work describes a novel codec for DNA storage called the Yin-Yang Codec (YYC). It's based on the observation that it is difficult to achieve both theoretical

information densities and sequences that meet critical biomolecular constraints, such as avoiding homopolymers, imbalanced GC content, or harmful structures. The YYC is effective at reaching high density because it supports a variety of encoding strategies.

Here's a brief summary of how it works. A file is divided into segments of a fixed length as binary strings; an index is added to each segment to mark its location in the file. Then, two segments, say a and b, are selected at random and sent through a two-step encoding process (Yin and Yang). One bit is taken from each of the two segments and is encoded into a single nucleotide. Using 2 bits to encode each nucleotide is what give it a high information density. The Yin step uses $a[i]$ to select a set of possible encoding bases, which narrows it down to 2 of the 4 bases based on a look-up table, call this set A. The Yang step uses the $b[i]$ bit and the previous nucleotide that was encoded to look up its encoding, which is also a set of possible nucleotides, lets call it set B. Then, the intersection of A and B is emitted as the output nucleotide. The Yin and Yang coding tables are designed to ensure a unique nucleotide is emitted. This process repeats for all bits in a and b until the sequence is generated. Because two bits are used to produce each base, it has the potential to reach the information theoretical limit of 2 bits / nt. However, after the sequence is created, it's sent through a filter to weed out bad sequences based on GC content, homopolymers, and thermodynamic analysis. If it fails this step, the sequence is thrown out. This process of selecting segments, encoding them, and checking them is repeated until enough good sequences are created to enable decoding of the file. No additional logical redundancy is included since this process is only meant as a way of encoding the data and is not meant to provide error correction.

Experiments are conducted in silico, in vitro, and in vivo to understand its utility and verify its usefulness for DNA storage. The YYC approach is applied to a suite of files of various formats and shown to successfully encode and decode while achieving good information densities. Its effectiveness is compared primarily to DNA Fountain given their overall similarity in goals and approach.

The most interesting part of the evaluation is in how well the codec works in silico. The in vitro studies are expected to work since they largely follow proven experimental methodologies. The in vivo studies add some new analysis to the field concerning storage and retrieval of data in yeast.

Significance

This work aims to further the field by presenting a new codec that provides high information density while still avoiding problematic sequences. This codec bears significant similarity to the goals of DNA Fountain, and that's why they compare against DNA Fountain. However, this work is not a Fountain code, and that enables some simplifications while also creating potential weaknesses.

Advantages

+A potential advantage of this work is the straightforward encoding and decoding process with respect to segments and their indices. This work can embed the indices directly in the encoded segments and recover them through standard decoding, but fountain codes need to encode that information as separate meta-data or hard code a lookup table or enumerator function a priori in the encoder and decoder.

+Another potential advantage of this work is the large variety of Yin and Yang coding tables, up to 1536, that allow a wide variety of ways to encode data. However, this may be a weakness, as described below.

Disadvantages:

- A fountain code can pick an arbitrary number of segments to merge, but this can only pick 2. YYC makes up for this limitation to some degree by having non-unitary mapping rules, however, this is achieved using a trial and error process. The trial and error process is not described in the paper or in the supplementary material, as far as I could tell. While the YYC can allow up to 1536 different coding tables, it was not demonstrated in the work that all of these combinations are actually useful, and I suspect that many of the tables end up producing similar results and may not all be equally useful.

Response 5: We thank the reviewer for the comprehensive summary and recognition of our work. To briefly summarize the two major concerns/suggestions from the reviewer: 1) providing the detailed description of the trial-and-error process; 2) describing the differences of the proposed 1536 coding schemes in this study. To answer the questions:

For the trial-and-error process, it is the iteration process as the reviewer understands in the above summary part: "This process of selecting segments, encoding them, and checking them is repeated until enough good sequences are created to enable decoding of the file". We further revised the description to make it clearer and highlighted it in the manuscript on page 5 (line 144 to 151).

For the difference analysis of 1,536 combinatory coding schemes, we followed the reviewer's suggestion and performed further analysis on the difference of generated DNA sequences by using individual coding scheme to transcode various types of files. The DNA sequence difference are evaluated by the average hamming distance, which is the number of bit positions in which the two bits are different. Our result shows that for the same digital files, 1328 of the 1536 coding schemes can generate corresponding DNA sequences with identity less than 40% and only less than 0.5% of full coding scheme collections (7 coding schemes) generating DNA sequences with identity between 80% to 91.85% (Fig. S3). **Our analysis suggests that the 1536 coding schemes can generate significantly different DNA sequences.** We have revised the manuscript on page 7 (line 192 to 196) and supplementary information on Figure S3 to further clarify.

- Fountain codes and rateless codes, in general, can add an arbitrary number of additional symbols to the encoded file, but this work can pick from at most n^2 combinations of segments, where n is the number of segments in the file, giving it far fewer combinations to work with to find usable sequences than Fountain codes.

Response 6: We thank the reviewer for the comment. We understand that the reviewer's concern is that n^2 combinations of segments may not be sufficiently enough to generate valid DNA sequences. First, we would like to point out that in the general circumstance, rateless codes like LT codes, Raptor codes, etc., can generate an arbitrary number of packets from the information source and increase the redundancy correspondingly to find usable sequences. However, that doesn't apply if the digital information shows extreme data patterns. We have proved our point in our work by transcoding 5 different binary patterns in Table. S1 and the figures of 9 different national flags shown in Table. S6 of the supplementary information. For DNA Fountain, even with 300% redundancy being introduced, there is still no solution for successful transcoding.

We would also like to clarify that the maximum combinations of YYC can be much higher than n^2 because YYC allows the introduction of additional "pseudo" binary segments to avoid the situation that there is no desired combination for existing segments (as stated in the manuscript on page 6 from line 165 to 167). Thus, the actual maximum number of combinations of YYC is $n \cdot 2^k$ (k refers to the number of additionally added segments). The approach by incorporating

additional binary segments is essentially useful and important for digital file with extreme data patterns. For the same examples we mentioned above that DNA Fountain failed to provide solution, using YYC, we successfully transcoded the figure into DNA sequences with high biocompatibility to DNA synthesis and sequencing processes.

Although archiving of the source digital file can balance the data pattern, however, as we described in the highlighted main text on page 14 (line 402-407), it also brings significant challenges to data decoding because the errors introduced during DNA synthesis and sequencing will significantly affect the success rate of decoding. To summarize, **our result suggests that YYC shows superior performance on the general coding/decoding processes by the proposed combination strategy with no preference on particular data patterns.**

- This work shares the disadvantage of DNA Fountain of needing to verify a sequence after its encoded, which means that either the pair of segments or the mapping tables could have been a poor choice. This can lead to large encode-time overheads and many failed attempts when encoding strands. The number of trials used to encode data and its compute time is not reported, but it should be for a fair assessment of the work. The work does report a limit of 100 attempts per strand, but the total number of attempts while encoding a file is not reported.

Response 7: We thank the reviewer for the comment. We understand the reviewer concerns that whether the iteration cycle/attempts cost significant encoding-time overheads to generate valid DNA sequence.

We performed the evaluation of average iteration run required by transcoding ten different formats of files and have shown that the average number from 1 to 7 trails (Table S3). We further took the reviewer's suggestion and conducted further analysis by estimating the total number of trails while encoding the 1 GB data collection used in our study for in silico simulation. As shown in Fig. S4a, in general, 65% of the segment can be successfully incorporated with another segment with a single run of trail, and for files with balanced byte frequency (Fig. R1 shown below), the percentage further increase to ~77% (Fig. S4b). From our observation, less than 0.3% of segments requires more than 10 runs of trail. **Our results imply that for an arbitrary file, YYC encoding would not cost a large encode-time overheads.** We have revised the manuscript accordingly to include these analysis results in the main text highlighted on page 6 (line 164 – 172), as well as supplementary (Figure S4).

Figure R1.

Statistics of byte frequency of all the files used in the simulation. According to Li et al., 2005, green ones were considered as files with balanced byte frequency. (For review usage only)

- The policy for what to do when a sequence fails to encode is not described in much detail. I am concerned with the computational complexity of choosing a suitable table -- how many attempts are needed? Also, I'm concerned about what happens if there is no suitable encoding for some part of the data. If 100 attempts are made and all fail, at this point, is a suitable synthetic sequence created just to

ensure encodability? Is this synthetic sequence somehow always guaranteed to succeed or might multiple attempts be needed here as well? It would be good to quantify this as well.

Response 8: We thank the reviewer for the comment. We think there might be some misunderstanding on that poor choice is made when suitable encoding failed after 100 attempts. **In the transcoding process of YYC, all the generated sequence must fulfill the constraints without any exception, and this is achieved by the incorporation of “pseudo” binary segment with random 0/1 but in balanced ratio as we mentioned in the response #6.** This can make sure that a valid DNA sequence will be generated for each binary segment. The addition of “pseudo” segment will of course reduce the bit-to-base information density. But according to our simulation analysis using the 1GB data collection, less than 0.002% of overall segments need this “back-up” plan (Fig. S4). In the “worst” case we observed in Table S3, the additional information added to the source file for successful transcoding accounts for only 19.25% of the original file size, with the average number of trails at ~ 7 and information density at ~ 1.45 . We have updated the description in the manuscript accordingly on page 6 (line 164 to 172).

Clarity and Context

* I think the work does not position itself as well as it could with respect to prior codecs or with respect to DNA Fountain. Many different codecs have been proposed to deal with various challenges of DNA storage. The presentation of where this one fits in and how it could be used with other approaches could be done with greater clarity. For example, this is really just part of the codec and doesn't include any new support for error correction. Also, this article takes a fairly limited view of what is relevant and does not discuss deeply how other codes were designed and what their advantages and disadvantages are. Organick et al. [Nature Biotech 2018] deal with these same issues but in different ways. Also, DNA Fountain, and fountain codes in general, could be configured and tuned to work potentially much better than the DNA Fountain work that is compared against, but this is not done or considered. The claimed advantages of YYC over DNA Fountain may be true for the specific implementation of DNA Fountain that was compared against, but those advantages likely do not extend to an optimized Fountain code for data storage.

Response 9: We thank the reviewer for the comment. To briefly summarize, we think there are two issues from the reviewer's view: 1) Comparison between YYC and other coding schemes apart from DNA fountain; relationship with other efforts (such as error correction) been made in the field of DNA storage; 2) Comparison towards DNA Fountain is limited at specific implementation.

For the first concern, as we described in the Response #4, excepting DNA fountain, early established coding schemes apply a single fixed rule for bit-to-base transcoding, in which no screening process is used. On the contrary, DNA Fountain and YYC are the only two coding schemes that combine transcoding rules and screening as a whole process to make sure the generated DNA sequences can meet the biochemical constraints, such as GC content and maximum homopolymer length. Thus, we reason that direct mathematical analysis should be focused on the comparison between DNA Fountain and YYC due to the similarity of coding strategy between YYC and DNA Fountain. Both algorithms employ the strategy of incorporation of two or more binary segments and generate a corresponding DNA sequence/ information packets. However, we did include the functional comparison with all developed coding scheme by encoding 1GB data collection in our study for in silico simulation analysis. Relative result is described in the "General principle and features of the Yin-Yang codec" part. **From our point of view, the whole DNA storage process includes several function modules: bit-to-base encoding, error-correction, indices assignment, redundancy handling, etc. YYC is developed as a bit-to-base encoding algorithm for improving the practicality and robustness of DNA data storage.** It can be used separately or in corporation with other developed function modules. For example, in this study we used RS code to implement the function of error correction. Future efforts can be performed to incorporate more functionalities and we proposed some of the orientations in the discussion part in main text, page 15 (line 418-424).

For the second suggestion, the in silico simulation analysis and experimental validations were performed with recommended configuration and parameters of DNA Fountain claimed in their study. In addition, we agree with the reviewer that the configuration and parameter adjustment could improve the general performance of a coding scheme like DNA Fountain, but the improvement will be in a relatively limited range. As described in our manuscript, increasing logical redundancy could greatly improve the probability of successful decoding for all the coding schemes. However, we have proven that even with exceeding high redundancy at 300% (Table S6), for some digital files, DNA Fountain still failed to generate an encoding solution. And

this is caused by the fundamental limitation of DNA Fountain rather than the selection of configuration and parameters.

* The example of the Yin-Yang Codec given in the paper is hard to understand. I think you should move a similar example as in the video into the paper. Make it concrete so that readers can understand how one of the configurations actually works. You can always add more explanation on how to generalize it elsewhere. Without a good example, the paper is not self-contained and reviewers will need the supplementary material to make sense of the work, which is not ideal in my opinion.

Response 10: We thank the reviewer for the comment. We have revised the manuscript and provided a similar example as in the video to make the paper self-contained (page 16 from line 469 to 481).

- The way indices are assigned to segments is not explained in enough detail. Also, how segments are assigned to either the Yin or Yang rule is not explained. These are important details because they have a large bearing on the overall efficiency of the system. For example, suppose a file is partitioned into segments and each one is labeled with a unique binary index, of length k (2^k total segments are possible). Now, suppose these segments are selected at random and passed arbitrarily into either the Yin or Yang rule. This means that k nucleotides are used to represent the index and 4^k sequences are possible, but only 2^k total indices are available in the file. This implies that indices have at most a coding density of 1 bit/base. Large files will need large indices, which substantially cuts into the claimed high density of the system since oligo synthesis has relatively hard limits on synthesized strand length. Other codecs which do not use a binary index will have a significant advantage in overall capacity and density. It's possible that I've misunderstood some aspect of how indices are assigned and used in this codec, so it would be good for this aspect of the paper to be improved.

Response 11: We thank the reviewer for the comment. The reviewer suggested that: 1) Incorporation of index will decrease the coding density to 1 bit/base; 2) Indices for large file could be enormous and might further reduce the information density; 3) Other coding schemes avoiding the use of binary index have advantage in the overall capacity and density.

For the first suggestion, we think there might be some misunderstanding about indices **assignment**. As the reviewer concluded, if a file is partitioned into 2^k segments and each one is labeled with a unique binary index, the index length will be k . The index will be attached to the partitioned binary segment as a new segment (information + index) before encoding corporation. Therefore, for the k nucleotides representing index, they are the corporation product of two indices, which means each nucleotide will encode 2 bits of the indices' information at most rather than 1 bit/base. We understand that k nucleotide will give 4^k possible sequences, but **coding density refers to (total information / total base), rather than (total information / possible choices)**.

For the second suggestion, we agree with the reviewer that large files indeed need more indices only if the configuration and parameters remain defined. However, there are many alternative choices other than index itself for the application of DNA storage. For example, flanking region usually has a length of 16-20 nucleotides for PCR amplification. And according to Organick et al., 2018, random access of DNA storage using flanking region can be one of the examples to expand the indexing. In addition, as the DNA synthesis technology keeps advancing, the length of synthetic DNA can be further increased to maintain a relatively high level of information density. In addition, using artificial data coding chromosome for storage further demonstrate the feasibility of reducing the number of indices and leads to a significant increase of coding capacity.

For the third suggestion, **according to Shannon's information theory, there will be always overheads to record the addresses, or indices**. For electrical devices, the address is the physical position of the spot, while for DNA storage the address is the index sequence. DNA fountain does not directly use a binary index. However, recording random seeds for segment trace-back can be also considered as indexing. Therefore, binary indices, or other forms for recording address, are indispensable elements especially for massive data storage by DNA. And in our study, we have demonstrated that YYC shows superior performance of information density in both *in vitro* and *in vivo* storage.

Major Suggestions

1. The paper makes a strong claim about density that needs to be clarified. As mentioned earlier, the work does not explain well how indices are handled (as

mentioned previously in my review), but the indices appear to achieve no better than a binary encoding (1 bit /base) density. One implication of this is that a high-density encoding is somewhat reliant on having small indices, which implies small file sizes and small capacity overall. This is an undesirable result for what should be a dense medium. Another implication of this is that the total capacity of the system (total unique strands) appears to be lower than those systems that achieve ternary indices, for example. While the claimed figures of merit for density are high, they hide the fact that the total capacity of a pool of DNA may be severely negatively impacted by binary indices. I would like to see a better explanation of how indices are handled and the effect this has on pool capacity.

Response 12: We thank the reviewer for the comment. As described in response 11, **the coding density refers to (total information / total base), rather than (total information / possible choices)**. Therefore, if one nucleotide encodes two binary digits of two different indices, the information density is 2 bits/base. In previous studies including Erlich et al., 2017, the density can be calculated in different ways. The coding potential (or information density) usually refers to (information stored/bases used to store information only). The coding density (or net information density) refers to (information stored/total bases used for information, indices, error-correction and flanking region). The physical density is related to experimental validation and refers to (information stored/total mass of DNA used). In this work, we used the consistent definition to calculate the information density. We believed that the ‘pool capacity’ the reviewer mentioned might refers to coding density (or net information density). Based on this, we have provided the statement of how indices would affect pool capacity and give some possible solution. we have revised the effect of indices on pool capacity accordingly in the main text highlighted on page 6 (line 179 – 182).

2. The capabilities of the YYC codec with respect to the variety of mapping tables is not well justified. It’s not clear that the ability to select one of the 1536 configurations actually helps or not. The work claims to leverage a combinatoric scheme, but that is not demonstrated through an experiment, except to some degree in the small one that analyzes strands with large fractions of 0 or 1 to see if any mapping tables can encode them satisfactorily.

To really demonstrate the advantage of all these combinations, several things need to be shown. (1) That having a choice among mapping tables is helpful and leads

to less overhead as compared to DNA Fountain. (2) That the choice of mapping table can be made efficiently. It would be interesting to report the number of failed attempts at selecting segments and encoding them. (3) That the overhead of holding the choice of mapping table in meta-data is a reasonable cost, even if done electronically. If a mapping table selection needs to be remembered per strand, then that implies approximately 11 bits of overhead per segment-pair, which for an exabyte scale system would add up to a very large cost. If it only needs to be remembered per file, it's a lower meta-data cost but higher compute cost to find a mapping. These trade-offs need some further explanations.

If this analysis is prohibitive to conduct, then I would suggest softening the claims over how useful these tables are.

Response 13: We thank the reviewer for the instructive comment.

First of all, we would like to clarify that the 1536 coding schemes of YYC offer alternative choices for encoding process, meaning it is not mandatory requirement to use all coding schemes in one case, and therefore the 11 bits of overhead is not for each segment-pair. We have demonstrated in the Response #5 that the 1536 coding schemes can generate significantly different DNA sequences. Thus, it offers one of the advantages that for an individual arbitrary file, we can always find some scheme(s) that can generate higher information density than others (Table S4). We performed a benchmarking test by encoding different files using all the 1,536 coding schemes. As described in the Response #7, we counted the iteration runs of these coding schemes on encoding 1 GB of data by all the 1,536 coding schemes. The incorporation failure rate for segment pairing over 100 iteration runs is only at 0.002%. In our previous study, we also performed some benchmarking tests on encoding and decoding efficiency (<https://doi.org/10.1101/2020.01.02.892588>), showing that the transcoding overheads of YYC is less than that of DNA fountain, YYC is 2-7 times faster.

For the efficiency of trail-and-error iteration, as we described in the Response #5 and 7, we performed the evaluation of average iteration run required by transcoding ten different formats of files and have shown that the average number from 1 to 7 trails (Table S3). We further took the reviewer's suggestion and conducted further analysis by estimating the total number of trails while encoding the 1 GB data collection used in our study for in silico simulation. As shown in Fig. S4a, in general, 65% of the segment can be successfully incorporated with another segment with a single run of trail, and for files with balanced byte frequency, the percentage further increase to

~77% (Fig. S4b). From our observation, less than 0.3% of segments requires more than 10 runs of trail. **Our results imply that for an arbitrary file, YYC encoding would not cost a large encode-time overheads.**

3. It's surprising that DNA Fountain does so poorly in the in silico experiment (Figure 2). I agree with the general arguments that DNA Fountain will have high degree packets that include many segments. However, I'm not convinced this is the entire answer unless DNA Fountain mostly selects high degree packets. Also, some important information is missing in this analysis. It appears that no error correction is used. In that case, what is done with strands that have insertions, deletions, and substitutions? Without error correction support, such problems cannot even be detected much less corrected. So, are they treated as correct and allowed to pollute the file with errors? This raises questions about how the DNA Fountain approach is utilized. Is the DNA Fountain code base used for this analysis? If so, is it possible that something is going wrong in how it handles these errors? For example, since segments are more likely repeated in multiple high degree packets, it may be detecting errors and throwing them out, whereas YYC has no such detection capability. Instead, YYC may be making a best-effort attempt to keep the data. More details of how this analysis is conducted are needed to fully explain DNA Fountain's poorer behavior.

Response 14: We thank the reviewer for the comment. The reviewer suggests that DNA fountain may have no bias on generating high degree packets. For fountain code itself and balanced byte frequency data pattern, high degree packets and low degree packets should share equal odds to be selected. However, **our observation is that the screening step of DNA Fountain offers an increased chance to pass the screening for high degree packets.** We performed a few tests to encode different files using DNA Fountain with and without screening to estimate the corresponding ratio of high degree packets (as the figure R2 shown below). We used a bmp image (united nation flag) and its zip-compressed package as test file. For zip file, of which the byte frequency is balanced (upper), there was no significant difference with screening (middle). However, for the original bmp image itself, it turned out that among the encoded files with screening, **the degree of packets generated is significantly biased without screening.**

Figure R2. Distribution of degree of packets generated by DNA fountain with/without screening (For review usage only)

The second suggestion from the reviewer is that error-correction code should be used in the in-silico test. As described in the Response #9, error correction codes, such as Reed-Solomon code used in our study, can be used as one of the function modules in the whole DNA storage process to improve the robustness for all coding schemes, but the improvement is within limited range. The reason is that current error-correction code is capable of correcting substitution errors but not insertion and deletion errors. For almost all known coding schemes including DNA Fountain and YYC, the sequence with indels cannot be corrected and is discarded before decoding. This is also the reason why error-correction code cannot improve the robustness of DNA fountain significantly. In our experimental validation, we encoded files using the original package provided by Erlich et al. with error correction code applied. The

result is consistent with in-silico simulation and proved our point of view. We have revised accordingly in the manuscript on page 9 (line 250-258).

Minor Suggestions

1. Perhaps add a brief explanation for a few things in the thermodynamic screening, such as at what temperature was the simulated free energy screening conducted? And, what other parameters were used, if any? Why did you pick -30 kcal/mol as the cutoff?

Response 15: We thank the reviewer for the comment. The thermodynamic screening cut-off is defined based on previous studies (Noguera et. al., 2014 and Yilmaz et. al., 2004). The free energy cutoff for probe design was set as -13 kcal/mol for ~20 nt DNA sequence, and considering the length of data coding DNA at 160nt, we adjusted the cutoff to – 30 kcal/mol. We have revised the manuscript accordingly to further clarify on page 18 (line 515 to 517).

2. In the text regarding in vitro experiments, please make it clear which encoding was used for each pool. I think P2 and P3 were done using DNA Fountain.

Response 16: We thank the reviewer for the comment. we have revised this accordingly in the main text highlighted on page 10 (line 296 – 302).

3. On Page 9, line 242-244, it says that loss of one strand in YYC can lead to the loss of two segments. This is true. But, isn't it possible that a segment is repeatedly selected for pairing due to some good pattern in its data, decreasing the odds it is lost? Or, do you prevent selecting the same sequence many times?

Thank you for an interesting paper.

Response 17: We thank the reviewer for the instructive comment and recognition of our work. We agree with the reviewer that repeated priority selection of segment with “good” pattern will increase the success rate of pairing. However, it will decrease the overall bit-to-base information density. In our study, we select each segment once and only once for the segment pairing process. The key difference between YYC and DNA Fountain is the segment incorporation strategy applied. For YYC each incorporation only involves two segments, so the loss of one segment will only affect the corresponding paired segment. However, for DNA Fountain the loss of one

segment will affect all related paired segment. **We have proven that without repeated selection of “good” pattern segments, YYC can greatly decrease the decoding failure caused by segment loss comparing to DNA Fountain.**

Decision Letter, second revision:

Date: 8th February 22 17:18:31
Last Sent: 8th February 22 17:18:31
Triggered By: Ananya Rastogi
From: ananya.rastogi@nature.com
To: shenyue@genomics.cn
CC: computationalscience@nature.com
Subject: AIP Decision on Manuscript NATCOMPUTSCI-21-0438B
Message: Our ref: NATCOMPUTSCI-21-0438B

8th February 2022

Dear Dr. Shen,

Thank you for submitting your revised manuscript "Towards Practical and Robust DNA-Based Data Archiving Using 'Yin-Yang Codec' System" (NATCOMPUTSCI-21-0438B). It has now been seen by the original referees and their comments are below. The reviewers find that the paper has improved in revision, and therefore we'll be happy in principle to publish it in Nature Computational Science, pending minor revisions to satisfy the referees' final requests and to comply with our editorial and formatting guidelines.

TRANSPARENT PEER REVIEW

Nature Computational Science offers a transparent peer review option for new original research manuscripts submitted from 17th February 2021. We encourage increased transparency in peer review by publishing the reviewer comments, author rebuttal letters and editorial decision letters if the authors agree. Such peer review material is made available as a supplementary peer review file. **Please state in the cover letter 'I wish to participate in transparent peer review' if you want to opt in, or 'I do not wish to participate in transparent peer review' if you don't.** Failure to state your preference will result in delays in accepting your manuscript for publication. Please note: we allow redactions to authors' rebuttal and reviewer comments in the interest of confidentiality. If you are concerned about the release of confidential data,

please let us know specifically what information you would like to have removed. Please note that we cannot incorporate redactions for any other reasons. Reviewer names will be published in the peer review files if the reviewer signed the comments to authors, or if reviewers explicitly agree to release their name. For more information, please refer to our [FAQ page](https://www.nature.com/documents/nr-transparent-peer-review.pdf).

Thank you again for your interest in Nature Computational Science Please do not hesitate to contact me if you have any questions.

Sincerely,

Ananya Rastogi, PhD
Associate Editor
Nature Computational Science

ORCID

Reviewer #3 (Remarks to the Author):

The authors addressed all my comments and the paper can be accepted.

Reviewer #4 (Remarks to the Author):

Thank you very much for the revisions and responses. You included some very nice analysis. Overall, they satisfy my concerns. I recommend acceptance.

I think there may be yet some misunderstanding between us about how indices are represented. I agree that you encode 2 bits per base for the indices. But, your approach appears to limit the total number of possible indices that fit into those k bases (k bases used for the index). This concerns me because it architecturally limits the pool capacity of the system. By pool capacity, I'm referring to the total amount of data in bytes that can be stored in the archive if this technique is employed. However, even if what I say is true, as you point out in your response, you can make up for it elsewhere with flanking primers, etc. So, I'm fine to accept the paper.

Author Rebuttal, second revision:

Reviewer's comments:

Reviewer #4 (Remarks to the Author):

Thank you very much for the revisions and responses. You included some very nice analysis. Overall, they satisfy my concerns. I recommend acceptance.

I think there may be yet some misunderstanding between us about how indices are represented. I agree that you encode 2 bits per base for the indices. But, your approach appears to limit the total number of possible indices that fit into those k bases (k bases used for the index). This concerns me because it architecturally limits the pool capacity of the system. By pool capacity, I'm referring to the total amount of data in bytes that can be stored in the archive if this technique is employed. However, even if what I say is true, as you point out in your response, you can make up for it elsewhere with flanking primers, etc. So, I'm fine to accept the paper.

Response 1: We thank the reviewer for the significant help in improving our work. It is true that large files indeed need more indices, but there are many other approaches can be applied to expand the indexing. As we mentioned in our previous point-by-point reply, the application of flanking primer is just one of the choices. Also, as relative technologies keep advancing, increasing the length of DNA that encodes digital data can actually decrease the need of total indices number. Due to the word limit, we include a brief discussion in the Discussion section in the main text as highlighted at page 10, line 315-318.

Final Decision Letter:**Date:** 18th March 22 17:19:05**Last Sent:** 18th March 22 17:19:05**Triggered By:** Ananya Rastogi**From:** ananya.rastogi@nature.com**To:** shenyue@genomics.cn**Subject:** Decision on Nature Computational Science manuscript NATCOMPUTSCI-21-0438C**Message:** Dear Dr Shen,

We are pleased to inform you that your Article "Towards Practical and Robust DNA-Based Data Archiving Using 'Yin-Yang Codec' System" has now been accepted for publication in Nature Computational Science.

Please note that *Nature Computational Science* is a Transformative Journal (TJ). Authors may publish their research with us through the traditional subscription access route or make their paper immediately open access through payment of an article-processing charge (APC). Authors will not be required to make a final decision about access to their article until it has been accepted. [Find out more about Transformative Journals](https://www.springernature.com/gp/open-research/transformative-journals)

Acceptance of your manuscript is conditional on all authors' agreement with our publication policies (see <https://www.nature.com/natcomputsci/for-authors>). In

particular your manuscript must not be published elsewhere and there must be no announcement of the work to any media outlet until the publication date (the day on which it is uploaded onto our web site).

Before your manuscript is typeset, we will edit the text to ensure it is intelligible to our wide readership and conforms to house style. We look particularly carefully at the titles of all papers to ensure that they are relatively brief and understandable.

Once your manuscript is typeset and you have completed the appropriate grant of rights, you will receive a link to your electronic proof via email with a request to make any corrections within 48 hours. If, when you receive your proof, you cannot meet this deadline, please inform us at rjsproduction@springernature.com immediately.

If you have queries at any point during the production process then please contact the production team at rjsproduction@springernature.com. Once your paper has been scheduled for online publication, the Nature press office will be in touch to confirm the details.

Content is published online weekly on Mondays and Thursdays, and the embargo is set at 16:00 London time (GMT)/11:00 am US Eastern time (EST) on the day of publication. If you need to know the exact publication date or when the news embargo will be lifted, please contact our press office after you have submitted your proof corrections. Now is the time to inform your Public Relations or Press Office about your paper, as they might be interested in promoting its publication. This will allow them time to prepare an accurate and satisfactory press release. Include your manuscript tracking number NATCOMPUTSCI-21-0438C and the name of the journal, which they will need when they contact our office.

About one week before your paper is published online, we shall be distributing a press release to news organizations worldwide, which may include details of your work. We are happy for your institution or funding agency to prepare its own press release, but it must mention the embargo date and Nature Computational Science. Our Press Office will contact you closer to the time of publication, but if you or your Press Office have any inquiries in the meantime, please contact press@nature.com.

We welcome the submission of potential cover material (including a short caption of around 40 words) related to your manuscript; suggestions should be sent to Nature Computational Science as electronic files (the image should be 300 dpi at 210 x 297 mm in either TIFF or JPEG format). We also welcome suggestions for the Hero Image, which appears at the top of our <http://www.nature.com/natcomputsci> home page; these should be 72 dpi at 1400 x 400 pixels in JPEG format. Please note that such pictures should be selected more for their aesthetic appeal than for their scientific content, and that colour images work better than black and white or grayscale images. Please do not try to design a cover with the Nature Computational Science logo etc., and please do not

submit composites of images related to your work. I am sure you will understand that we cannot make any promise as to whether any of your suggestions might be selected for the cover of the journal.

Best regards,

Ananya Rastogi, PhD
Associate Editor
Nature Computational Science

P.S. Click on the following link if you would like to recommend Nature Computational Science to your librarian: https://www.springernature.com/gp/librarians/recommend-to-your-library

** Visit the Springer Nature Editorial and Publishing website at www.springernature.com/editorial-and-publishing-jobs for more information about our career opportunities. If you have any questions please click here. **